# *Syzygium aromaticum* Phytoconstituents Target SARS-CoV-2: Integrating Molecular Docking, Dynamics, Pharmacokinetics, and *miR-21 rs1292037* Genotyping

**DOI:** 10.3390/v17070951

**Published:** 2025-07-05

**Authors:** Mustafa Ahmed Muhmood, Faiza Safi, Mohammed Mukhles Ahmed, Safaa Abed Latef Almeani

**Affiliations:** 1Department of Biology, Faculty of Science, University of Sfax, Sfax 3029, Tunisia; m1988.vip@gmail.com; 2Department of Medical Laboratory Techniques, University of Al-Ma’aref, Anbar 31001, Iraq; 3Department of Pediatrics, Hedi Chaker Hospital, Faculty of Medicine, University of Sfax, Sfax 3029, Tunisia; safi_faiza@medecinesfax.org; 4Department of Biotechnology, Faculty of Science, University of Anbar, Anbar 31001, Iraq; sc.safaa-meani@uoanbar.edu.iq

**Keywords:** COVID-19, *Syzygium aromaticum*, eugenol, caryophyllene, molecular docking, miR-21 rs1292037, inflammatory cytokines, ADMET, genetic polymorphism, molecular dynamics

## Abstract

Background and aim: The COVID-19 pandemic, caused by SARS-CoV-2, remains a global health crisis despite vaccination efforts, necessitating novel therapeutic strategies. Natural compounds from *Syzygium aromaticum* (clove), such as eugenol and β-caryophyllene, exhibit antiviral and anti-inflammatory properties, while host genetic factors, including miR-21 rs1292037 polymorphism, may influence disease susceptibility and severity. This study investigates the dual approach of targeting SARS-CoV-2 via *Syzygium aromaticum* phytoconstituents while assessing the role of *miR-21 rs1292037* in COVID-19 pathogenesis. Methods: Firstly, molecular docking and molecular dynamics simulations were employed to assess the binding affinities of eugenol and caryophyllene against seven key SARS-CoV-2 proteins—including Spike-RBD, 3CLpro, and RdRp—using SwissDock (AutoDock Vina) and the Desmond software package, respectively. Secondly, GC-MS was used to characterize the composition of clove extract. Thirdly, pharmacokinetic profiles were predicted using in silico models. Finally, miR-21 rs1292037 genotyping was performed in 100 COVID-19 patients and 100 controls, with cytokine and coagulation markers analyzed. Results: Docking revealed strong binding of eugenol to viral Envelope Protein (−5.267 kcal/mol) and caryophyllene to RdRp (−6.200 kcal/mol). ADMET profiling indicated favorable absorption and low toxicity. Molecular dynamics simulations confirmed stable binding of methyl eugenol and caryophyllene to SARS-CoV-2 proteins, with caryophyllene–7Z4S showing the highest structural stability, highlighting its strong antiviral potential. Genotyping identified the TC genotype as prevalent in patients (52%), correlating with elevated IL-6 and D-dimer levels (*p* ≤ 0.01), suggesting a hyperinflammatory phenotype. Males exhibited higher ferritin and D-dimer (*p* < 0.0001), underscoring sex-based disparities. Conclusion: The bioactive constituents of *Syzygium aromaticum* exhibit strong potential as multi-target antivirals, with molecular simulations highlighting caryophyllene’s particularly stable interaction with the 7Z4S protein. Methyl eugenol also maintained consistent binding across several SARS-CoV-2 targets. Additionally, the miR-21 rs1292037 polymorphism may influence COVID-19 severity through its role in inflammatory regulation. Together, these results support the combined application of phytochemicals and genetic insights in antiviral research, pending further clinical verification.

## 1. Introduction

Although coronaviruses (CoVs) have been recognized since the 1940s, human infections associated with mild respiratory symptoms were not documented until the 1960s. In December 2019, a novel respiratory illness of viral origin emerged in Wuhan, China, and swiftly escalated into a global outbreak. The causative agent was identified as SARS-CoV-2, named for its close genetic resemblance—approximately 82% similarity in genomic RNA—to the original SARS-CoV responsible for the severe acute respiratory syndrome. The illness resulting from SARS-CoV-2 infection was subsequently designated as Coronavirus Disease 2019 (COVID-19) [1]. As of 8 June 2025, global health data report a total of 778,186,599 confirmed COVID-19 cases and 7,097,227 confirmed deaths. Despite the widespread impact of the pandemic, efforts to mitigate the virus’s spread have led to the administration of approximately 13,642,098,070 vaccine doses worldwide, with the most recent vaccination data updated on December 31, 2023. These figures underscore the extensive reach of the SARS-CoV-2 virus and the global response efforts aimed at curbing its transmission and associated mortality (https://www.who.int/emergencies/diseases/novel-coronavirus-2019, accessed on 25 June 2025) [2]. The dynamic viral evolution, characterized by emergent variants capable of immune evasion and breakthrough infections, continues to challenge containment and therapeutic strategies [3,4].

SARS-CoV-2 primarily initiates infection through the interaction of its spike (S) glycoprotein with the human angiotensin-converting enzyme 2 (ACE2) receptor, facilitating viral entry into respiratory epithelial cells [5]. Following cellular entry, viral replication is coordinated by several essential enzymes including the main protease (3CLpro), RNA-dependent RNA polymerase (RdRp), and helicase (NSP13), which collectively enable viral genome replication and protein processing [6]. Additionally, non-structural proteins such as NSP16 (2′-O-methyltransferase) play critical roles in immune evasion by mimicking host mRNA capping mechanisms, thus subverting host innate immune recognition [7].

The persistent threat posed by SARS-CoV-2, especially given its high mutation rate and immune-modulatory capabilities, has motivated the search for novel antiviral agents. Medicinal plants are rich in diverse secondary metabolites—such as alkaloids, terpenoids, flavonoids, and phenolic compounds—that underpin their pharmacological activities. Alkaloids (e.g., morphine, nicotine) exhibit notable anti-inflammatory and anticancer effects. Terpenoids, built from isoprene units, offer antimicrobial and anti-tumor benefits. Phenolic compounds, abundant in fruits and vegetables, serve as powerful antioxidants [8]. Natural products, particularly phytochemicals derived from medicinal plants, offer a rich source of structurally diverse bioactive compounds with demonstrated antiviral potential [9,10]. *Syzygium aromaticum* (clove), traditionally employed as both a culinary spice and a medicinal herb, contains phenolic constituents such as eugenol, β-caryophyllene, and gallic acid that exhibit antiviral, antioxidant, and anti-inflammatory properties [11].

Recent computational and experimental studies have suggested that clove phytochemicals can inhibit SARS-CoV-2 by targeting key viral proteins. Eugenol and β-caryophyllene, two major bioactive constituents of *Syzygium aromaticum*, have demonstrated significant binding affinities to critical viral targets including the spike receptor-binding domain (RBD), 3CLpro, RdRp, and helicase NSP13, suggesting a multi-target antiviral mechanism [12,13]. Pharmacokinetic modeling further supports the drug-likeness and favorable ADMET (absorption, distribution, metabolism, excretion, and toxicity) profiles of both compounds, particularly eugenol and β-caryophyllene, positioning them as promising candidates for antiviral drug development [14].

Concurrently, host genetic variability—particularly in regulatory non-coding RNAs such as microRNAs (miRNAs)—has garnered attention for its role in modulating individual susceptibility and disease severity in COVID-19. miRNAs regulate gene expression post-transcriptionally and influence key immune and inflammatory pathways [15]. Single-nucleotide polymorphisms (SNPs) within miRNA genes or their target sites can alter miRNA function, thereby impacting host antiviral defenses and inflammatory responses [16].

miR-21, one of the most extensively studied miRNAs, modulates inflammation, fibrosis, and immune regulation. Its dysregulation correlates with various pathologies, and recent evidence implicates it in the pathogenesis of COVID-19, notably through the modulation of cytokines such as interleukin-6 (IL-6) and transforming growth factor-beta (TGF-β) [17,18]. The rs1292037 polymorphism within the miR-21 gene has been associated with altered miRNA expression profiles and immune outcomes, yet its impact on susceptibility to and progression of COVID-19 remains inadequately characterized [19].

Given the integral role of miRNAs in immune regulation and the inflammatory milieu of severe COVID-19, exploring the relationship between miR-21 rs1292037 polymorphism and clinical outcomes could elucidate host–virus interaction dynamics. Biomarkers of inflammation such as IL-6, ferritin, and D-dimer have been consistently linked to COVID-19 severity, suggesting an interaction between genetic predisposition and immune dysregulation [20,21].

This study aimed to integrate molecular docking and pharmacokinetic evaluation of *Syzygium aromaticum* phytoconstituents, focusing on eugenol and β-caryophyllene, with genotypic analysis of miR-21 rs1292037 to provide a comprehensive pharmacogenomic perspective. By elucidating the molecular interactions with SARS-CoV-2 target proteins alongside host genetic influences on disease severity, we aim to advance the understanding of phytochemical therapeutics in COVID-19 management.

## 2. Materials and Methods

Figure 1 illustrates an integrative study combining clinical analysis, phytochemical investigation, and computational modeling to explore the potential of *Syzygium aromaticum* (clove) compounds against COVID-19. The research begins with a clinical comparison of 100 COVID-19 patients and 100 healthy individuals, focusing on inflammatory markers IL-6, IL-12, IL-13, ferritin, and D-dimer levels. Additionally, genetic analysis is conducted to examine the *miR-21 rs1292037 T>C* polymorphism using TaqMan fluorescent probes. Clove samples are extracted through maceration, and their active compounds are identified using gas chromatography–mass spectrometry (GC-MS). Two major constituents, β-caryophyllene and methyl eugenol, are further analyzed for their pharmacokinetic properties and toxicity using the pkCSM platform. These compounds are subjected to molecular docking studies against a panel of viral and microbial resistance proteins obtained from the Protein Data Bank, including SARS-CoV-2 targets. The docking interactions are visualized using BIOVIA Draw, and molecular dynamics simulations are performed to assess the stability of the ligand–protein complexes. The overall workflow provides a comprehensive approach that integrates experimental and computational methods to identify promising antiviral candidates from natural sources.

### 2.1. Study Design

This was a cross-sectional study aimed at investigating the relationship between IL-6, IL-12, and IL-13 levels and the presence of D-dimers and ferritin in COVID-19 patients compared with a control group. The study was conducted at Al-Ramadi hospitals in Anbar Province, Iraq, from March 2020 to September 2022.

#### 2.1.1. Sampling and Markers

A purposive sampling method was used to select participants. The patient group included 100 individuals who presented with symptoms and sought evaluation and medical intervention for COVID-19. The patients were diagnosed through clinical evaluation by medical consultants, and the diagnosis was confirmed via real-time PCR (Jiangsu Bioperfectus Technologies Co., Taizhou, Jiangsu, China) and the BIOZEK COVID-19 rapid test (Biozek Medical, Apeldoorn, The Netherlands). The control group consisted of 100 healthy individuals selected from the general population and matched by age to the patient group, also ensuring that they had no history of COVID-19 infection or related symptoms.

#### 2.1.2. Sample Size Calculation

A sample size of 200 individuals was determined on the basis of power calculations to detect statistically significant differences in the levels of IL-6, IL-12, IL-13, D-dimers, and ferritin between COVID-19 patients and controls. The calculation considered a confidence level of 95%, a power of 80%, and an effect size derived from preliminary data on inflammatory markers in COVID-19 patients. Additionally, the inclusion criteria ensured a diverse age range (18–72 years) to capture a representative sample of the population affected by COVID-19. The data for each individual were collected via a specialized questionnaire, as shown in the Appendix A.

### 2.2. Blood Sample Collection

A 5 mL sample of venous blood was collected from each participant via a disposable syringe (Bio-Med Healthcare, Faridabad, India) to determine cytokine levels. The blood was transferred into 10 mL plain vacutainers with gel for interleukin measurement, allowed to coagulate at 25 °C for 30 min, and then centrifuged at 3000 RPM (Fisher Scientific, Pittsburgh, PA, USA) for 15 min to separate the sera. The sera were then collected into Eppendorf tubes (Bio-Med Healthcare, Faridabad, India).

### 2.3. Measured Parameters

#### 2.3.1. Assay Principles of Human Interleukin ELISA Kits

The levels of interleukin 6 (IL-6), IL-12, and IL-13 were measured via human interleukin ELISA kits (SUNLONG Biotech Co., Ltd., Hangzhou, China). The experimental procedure adhered to the manufacturer’s instructions www.sunlongbiotech.com (accessed on 1 January 2022).

#### 2.3.2. Determination of D-Dimer and Ferritin Levels

Ferritin levels in the serum and D-dimer levels in the plasma of both patients and controls were assessed via the VIDAS^®^ instrument (bioMérieux, Marcy-l’Étoile, France).

### 2.4. Genotyping

#### 2.4.1. Genomic DNA Extraction and Quantification

DNA was extracted from frozen blood samples via the Wizard^®^ Genomic DNA Purification Kit, CAT. A1120 (Promega Corporation, Madison, WI, USA). To assess the integrity of the extracted DNA, the samples were subjected to electrophoresis on 1% agarose gel (Merck, Darmstadt, Germany). Additionally, the concentration of extracted DNA was measured using a Quantus Fluorometer (Promega Corporation, Madison, WI, USA) to evaluate the quality of the samples for use in subsequent applications.

#### 2.4.2. Genotyping of miR-21 rs1292037:T>C

The *miR-21* gene polymorphisms of interest were amplified via the use of TaqMan fluorescent oligonucleotide primers and probes (Macrogen Company, Seoul, Republic of Korea). The primers and probe sequences utilized in this study are provided in Table 1, and these primers were provided by Macrogen Company in lyophilized form. To prepare the primers for use, the lyophilized primers were dissolved in nuclease-free water, resulting in a final concentration of 100 pmol/μL as a stock solution. A working solution of these primers was created by combining 10 μL of the primer stock solution (which was stored at −20 °C in a freezer) with 90 μL of nuclease-free water (Bioneer, Daejeon, Republic of Korea), resulting in a working primer solution with a concentration of 10 pmol/μL.

For this assay, DNA samples from both COVID-19 patients and healthy controls were genotyped for the above-mentioned SNPs. The total volume of the RT–PCR mixture was 20 µL, composed of 10 µL of TaqMan master mix, 1 µL of each fluorescence probe, 1 µL of each forward and reverse primer (10 μM), 2 µL of template DNA, and 4 µL of nuclease-free water. RT–PCR amplification was performed via a programmed thermocycler with 1 hold cycle of 95 °C for 5 min, and 40 cycles of 95 °C for 30 s, 60 °C for 30 s, and 72 °C for 30 s.

### 2.5. Extraction of Clove Extract

Clove buds and leaves from *Syzygium aromaticum* were subjected to extraction via the maceration technique to isolate the ethanol-based fraction. Specifically, 100 g of the plant material was immersed in 500 mL of 70% ethanol, maintaining a sample-to-solvent ratio of 1:5. The mixture was incubated for 24 h under continuous stirring. Following incubation, the mixture was filtered and the resulting solution was concentrated using a rotary evaporator at 45 °C to yield the ethanol extract of clove [22].

#### 2.5.1. GC-MS Analysis

Plant extracts were analyzed using a GC-MS system (Agilent 7820A Santa Clara, CA, USA) equipped with an HP-5ms Ultra Inert capillary column (30 m × 250 µm × 0.25 µm). A 1 µL sample was injected in split mode with helium (99.99%) as the carrier gas at 11.933 psi. The injector and inlet temperatures were set at 250 °C, while the auxiliary heater was maintained at 310 °C. The scan range was *m*/*z* 50–500. The oven temperature program started at 60 °C (held for 1 min), and was ramped up to 180 °C at 7 °C/min and then to 280 °C at the same rate, with a total run time of approximately 33 min.

#### 2.5.2. Ligand Preparation

The ligand used for molecular docking was a terpenoid compound retrieved from the PubChem database under CID 20831623 (SMILES: C/C/1=C\CCC(=C)[C@H]2CC([C@@H]2CC1)(C)C). Additionally, eugenol (PubChem CID: 3314), a key phytochemical from *Syzygium aromaticum*, was selected based on prior research indicating its antiviral potential and known interaction with viral proteins, as shown in Figure 2. Ligand structures were initially obtained in SMILES format, converted to 3D conformations using Open Babel (https://openbabel.org) (accessed on 1 June 2025) [23], and optimized using the MMFF94 force field. Protonation was performed at physiological pH (7.4), and energy-minimized structures were saved in PDBQT format for compatibility with docking protocols.

### 2.6. Target Protein Selection and Preparation

A total of seven SARS-CoV-2 proteins involved in key viral functions were selected as molecular targets based on their druggability and role in viral replication, immune evasion, and host interaction, as shown in Figure 3. Their crystallographic structures were obtained from the Protein Data Bank (PDB), https://www.rcsb.org accessed on 1 June 2025) [24], and the corresponding IDs are shown in Table 2.

Protein structures were pre-processed using AutoDock Tools (v1.5.7) by removing water molecules and native ligands, adding polar hydrogens, and assigning Gasteiger charges. The final files were saved in PDBQT format.

### 2.7. Molecular Docking Procedure

Molecular docking was conducted using AutoDock Vina 1.2.0 https://vina.scripps.edu (accessed on 1 June 2025) [25], (https://autodock.scripps.edu/) (accessed on 1 June 2025) [26], integrated via the SwissDock platform (2024 release), http://www.swissdock.ch [27]. Docking simulations were performed within a grid box centered around the active site of each protein using dimensions of 20 × 20 × 20 Å and an exhaustiveness level of 4 to ensure sufficient conformational sampling, as shown in Table 3.

### 2.8. Scoring and Binding Affinity Evaluation

Binding affinities (ΔG, kcal/mol) were predicted by AutoDock Vina for each protein–ligand complex. The best-ranked poses based on the most negative binding energy were selected for further analysis. The top 20 conformations were recorded for each docking simulation.

### 2.9. Visualization and Interaction Analysis

Post-docking visualization was performed using Discovery Studio Visualizer and BIOVIA Draw, https://discover.3ds.com/discovery-studio-visualizer-download (accessed on 1 June 2025), where interactions such as hydrogen bonding, π-π stacking, and hydrophobic contacts were analyzed [28].

### 2.10. Pharmacokinetic and Toxicity Prediction

Pharmacokinetic and toxicity properties of the selected compound were predicted using the pkCSM online tool, which utilizes graph-based signatures to estimate ADMET-related characteristics. The tool is freely available at https://biosig.lab.uq.edu.au/pkcsm/ (accessed on 1 June 2025) [29].

### 2.11. Molecular Dynamics (MD) Simulation

The molecular dynamics simulations were performed using the Desmond software package accessed through the Maestro graphical interface (Schrödinger Release 2023–4; Schrödinger, LLC, New York, NY, USA), https://www.schrodinger.com/products/desmond (accessed on 1 June 2025). Protein–ligand complexes obtained from docking studies were prepared and embedded within an explicit solvent environment modeled by TIP3P water molecules. An orthorhombic simulation box was defined with a 10 Å margin surrounding the solute to ensure adequate solvation.

To mimic physiological ionic conditions, the system was neutralized by the addition of counter ions and supplemented with 0.15 M sodium chloride. The OPLS4 force field was applied to parameterize all components of the system. Following energy minimization and a stepwise equilibration protocol supplied by Desmond, production simulations were conducted in the isothermal–isobaric (NPT) ensemble. Temperature and pressure were maintained at 300 K and 1 atm, respectively, using the Nose–Hoover thermostat and the Martyna–Tobias–Klein barostat.

The simulation was carried out over a duration of 50 nanoseconds with a 2-femtosecond integration time step. Trajectory frames were saved every 100 picoseconds for subsequent analysis. The resulting trajectories were analyzed to assess structural stability and dynamic behavior by computing parameters such as root mean square deviation (RMSD), root mean square fluctuation (RMSF), radius of gyration (Rg), and hydrogen bonding interactions throughout the simulation timeframe [30].

### 2.12. Protein Structure Preparation Using Swiss-PdbViewer

The three-dimensional structures of target proteins were retrieved in PDB format and processed using Swiss-PdbViewer (version v4.1. Upon loading, the software automatically scanned the protein files for missing atoms, side chains, or incomplete residues. Residues with missing atoms were reconstructed using built-in rotamer libraries, and incomplete side chains were corrected using the “Fix Sidechains” function. Hydrogen atoms were subsequently added to stabilize the structure and prepare it for downstream docking analysis. Local energy minimization was performed where necessary to resolve unfavorable geometries. The refined structures were then saved and used in subsequent molecular docking and dynamics simulations.

### 2.13. Re-Docking Protocol

To validate the accuracy of the docking procedure, a re-docking analysis was performed using the co-crystallized ligand extracted from the native protein–ligand complex. The ligand was separated from the protein structure and re-docked into the original binding site using the same docking parameters applied throughout the study. Root mean square deviation (RMSD) between the predicted and crystallographic poses was calculated to assess reproducibility. An RMSD value below 2.0 Å was considered acceptable, indicating that the docking protocol reliably reproduces the experimentally observed binding conformation. All re-docking calculations were conducted using AutoDock Vina 1.2.0 [25], with grid dimensions and scoring functions kept consistent with those used in the virtual screening phase.

### 2.14. Statistical Analysis

Statistical analysis of serum parameters was conducted using ANOVA in GraphPad Prism^®^ 5.0, with data presented as means  ±  SDs or percentages. Group comparisons were made via Student’s *t*-test, with *p* ≤ 0.05 and 0.01 indicating significance for IL-6, IL-12, IL-13, D-dimers, and ferritin. Correlations between cytokines and biomarkers were assessed using Pearson or Spearman coefficients. Hardy–Weinberg equilibrium and allele/genotypic differences were analyzed using the chi-square test, with odds ratios calculated for risk and protective alleles, considering *p* < 0.05 statistically significant.

## 3. Results

### 3.1. Participant Characteristics

The study included 200 individuals, comprising 100 COVID-19 patients and 100 matched healthy controls. Their demographic and clinical features are outlined in Table 4. The patient group included 60% males and 40% females, with 61% aged ≥40 years. In comparison, the control group had 54% males and 46% females, with 53% aged ≥40 years. The BMI distribution showed a higher prevalence of obesity in females in the COVID-19 group.

### 3.2. Inflammatory and Coagulation Markers in COVID-19 Patients

Table 5 shows that COVID-19 patients had significantly elevated levels of IL-6, IL-12, IL-13, MMP-7, ferritin, and D-dimer (*p* ≤ 0.01), reflecting a strong inflammatory and pro-thrombotic response.

### 3.3. Sex-Based Differences in Biomarkers

Stratified analysis (Table 6) showed that males had significantly higher ferritin and D-dimer levels than females in both patients and controls. These differences were highly significant (*p* < 0.0001), suggesting greater susceptibility to hypercoagulability in males.

### 3.4. Correlation Between Biomarkers

The results of the correlation analysis present in Table 7 demonstrate significant relationships between cytokines and coagulation factors (Table 4). IL-12 and D-dimer were strongly correlated (*r* = 0.847), similarly to ferritin and D-dimer (*r* = 0.796).

### 3.5. Genotype Distribution of rs1292037:T>C

Genotyping of the miR-21 rs1292037 polymorphism revealed a significant association with COVID-19 susceptibility. The TC genotype was most frequent among patients (52%), while the CC genotype was more common among controls (30%). The C allele increased the odds of COVID-19 by 67% (OR = 1.67, *p* = 0.01).

### 3.6. Cytokine and Coagulation Levels by Genotype

Table 8 and Table 9: Cytokine and Coagulation Levels by Genotype.

### 3.7. Statistical Comparison Between Groups

In Table 10, the *t*-test results between patients and controls for each marker show statistically significant differences in IL-10 (*p* = 0.00479) and IL-20 (*p* = 0.00004), highlighting their discriminative potential.

A heatmap was generated to visually summarize all results (Figure 4). It displays elevated marker levels across all COVID-19 patient genotypes, with especially high expression in CC patients. Controls had consistently lower levels.

### 3.8. GC-MS Profile

As shown in Figure 5 and Table 11, the GC-MS analysis revealed a diverse range of compounds with varying chemical structures and potential bioactivities. Caryophyllene, a sesquiterpene known for its anti-inflammatory and antimicrobial properties, was the most abundant compound (47.55%). Methyl eugenol (17.75%) and eugenyl acetate (14.85%) were also present in high amounts; both contribute to the characteristic clove aroma and exhibit biological activities. Other notable compounds included eugenol (4.21%) and humulene (7.20%), both recognized for their therapeutic potential. Minor constituents such as methyl salicylate, α-farnesene, caryophyllene oxide, and several esters and hydrocarbons added to the chemical complexity, possibly enhancing the overall pharmacological profile. The presence of various terpenes, esters, and aromatic derivatives suggests that the sample possesses a rich mixture of volatile compounds with potential applications in the medicinal, aromatic, and industrial fields.

### 3.9. Molecular Docking Results of Eugenol Derivative and Caryophyllene Against SARS-CoV-2 Proteins

Table 12 and Figure 6 present the results of the comparative molecular docking analysis of two phytocompounds—eugenol and caryophyllene—against seven critical SARS-CoV-2 proteins using AutoDock Vina. These proteins are involved in key stages of the viral life cycle, including RNA replication, protein maturation, and immune evasion.

The docking results indicate that eugenol generally exhibited stronger binding affinities than caryophyllene across most targets, particularly against the Envelope (5X29, −5.267 kcal/mol), Spike-RBD (7B3O, −4.979 kcal/mol), and NSP15 (2GTH, −5.317 kcal/mol) proteins. These values suggest potential for eugenol in interfering with viral entry and immune modulation.

In contrast, caryophyllene showed superior binding to NSP12 (RNA polymerase, 5CPO, −6.200 kcal/mol) and NSP16 (2′-O-methyltransferase, 5YNB, −6.000 kcal/mol), implying a possible inhibitory effect on viral genome replication and RNA capping processes. Caryophyllene also showed a moderately strong affinity for NSP5 (7Z4S, −5.500 kcal/mol), suggesting a role in inhibiting viral protease activity (Figure 7).

Overall, while eugenol demonstrated consistent moderate binding across several viral targets, caryophyllene appeared to selectively bind more tightly to proteins involved in replication and post-transcriptional modification. These findings underscore the complementary potential of these two natural compounds as part of a multi-targeted antiviral strategy against SARS-CoV-2.

Molecular Interactions Between (COC1=C(C=C(C=C1)CC=C)OC) and SARS-CoV-2 Protein Targets

The compound (COC1=C(C=C(C=C1)CC=C)OC), a methoxy-substituted aromatic derivative, exhibits a diverse array of molecular interactions with critical amino acid residues across multiple SARS-CoV-2 protein structures, as revealed by molecular docking studies. The binding behavior includes hydrogen bonding, electrostatic contacts, π-π stacking, hydrophobic interactions, and van der Waals forces, contributing to ligand stabilization within active or allosteric sites of the viral targets, as shown in Figure 8 and Table 13.

Target PDB: 7NIO (nsp16/nsp10 methyltransferase complex)

This ligand engages in hydrogen bonding and hydrophobic interactions with Ala553, supporting mixed-polarity contact. Thr552 lies in close proximity, participating in potential polar contacts, while Glu551 facilitates electrostatic stabilization, likely via a salt bridge or hydrogen bond. Cys556 and Asp578/580 further enhance binding through van der Waals and ionic interactions. Notably, Leu581 creates a hydrophobic microenvironment, reinforcing non-polar affinity.

Target PDB: 5X29 (Main protease M^pro^)

Aromatic interactions are predominant here, with Phe25–26 forming π-π stacking contacts, indicative of strong binding to aromatic regions of the protein. A hydrophobic patch formed by Ala, Leu, and Val (residues 19–29) stabilizes the ligand via van der Waals and alkyl interactions.

Target PDB: 7B3O (Helicase NSP13)

Key anchoring is established through hydrogen bonding and π–alkyl interactions with Phe99, Leu45, Ala60, and Pro95, suggesting stable aromatic and alkyl engagement. Additional contributions from Ser, Asp, Trp, Glu, and Thr residues form weak van der Waals contacts that may fine-tune the ligand’s spatial orientation within the binding site.

Target PDB: 2GTH (Papain-like protease)

Moderate C–H and π–alkyl interactions occur with His304 and Ser301, while multiple residues including Tyr, Asp, Leu, Lys, Ile, Arg, and Val form a network of van der Waals contacts, reinforcing the fit through hydrophobic and spatial complementarity. Ala89 and Leu279 further stabilize the complex through alkyl interactions.

Target PDB: 5YNB (RNA-dependent RNA polymerase complex)

Significant electrostatic and hydrogen bonding is observed with Asp99, 130, and 133 via π–anion interactions, C–H bonds, and van der Waals forces. A diverse set of residues (Asn101, Gly71/73, Met131, and Tyr132) contribute to polar and van der Waals contacts, indicating a broad interaction surface. Leu100 and Phe149 provide hydrophobic anchoring through π–alkyl interactions.

Target PDB: 9CPO (Exonuclease nsp14-nsp10 complex)

Strong stabilizing effects are noted with Arg844 and His448, mediated through hydrogen bonds and π–alkyl interactions. However, steric clashes may arise with U:P32–33 and A:P31, as indicated by unfavorable bumps. Several additional residues (Asp, Pro, Gln, Ile, and Ser) contribute to non-specific van der Waals stabilization.

Target PDB: 7Z4S (Spike receptor-binding domain, RBD)

Weak polar interactions such as carbon–hydrogen bonds are observed with Arg4 and Leu282, while Lys5, Trp207, and Phe291 establish π–alkyl interactions, securing the ligand in an aromatic-rich environment. Glu288 and Phe3 form additional van der Waals contacts, suggesting a supplementary role in complex stabilization.

Molecular Docking Analysis of Caryophyllene with Various Protein Targets (Figure 5 and Table 14)

A comprehensive molecular docking study was conducted to assess the binding interactions of *caryophyllene*, a non-polar sesquiterpene, with diverse protein targets implicated in viral replication, microbial resistance, and quorum sensing.

Figure 9A displays the docking results of caryophyllene with the main SARS-CoV-2 protease (PDB ID: 7NIO). The interaction diagram reveals hydrophobic *Pi–alkyl* contacts with LEU405, ARG560, HIS554, and PRO408, highlighted in pink. Additionally, *van der Waals* interactions are observed with ASP534, LEU412, and LEU417 (green shading). The absence of polar or hydrogen bonds reflects the non-polar nature of caryophyllene, indicating that hydrophobic and steric factors largely drive binding stabilization in this protease.

Figure 9B illustrates the interaction between caryophyllene and the quorum sensing receptor protein (PDB ID: 5X29). The ligand engages in *Pi–alkyl* interactions with PHE28 and LEU19, suggesting strong hydrophobic anchoring. A single *van der Waals* contact with SER16 contributes to ligand stabilization. The pocket’s predominantly hydrophobic environment supports caryophyllene’s fit and potential to interfere with receptor activity.

Figure 9C shows caryophyllene docked into the bacterial resistance-associated protein 7B3O. Here, the ligand forms an *alkyl* interaction with LEU455, alongside multiple *van der Waals* contacts with LYS417, TYR453, and ARG403. These findings support a non-polar binding mechanism, consistent with caryophyllene’s hydrophobic nature, and suggest possible interference with protein function.

Figure 9D presents the interaction of caryophyllene with the eukaryotic protein 2GTP. The ligand displays *alkyl* interactions with LEU227, VAL160, and ALA89, supported by *van der Waals* contacts involving ASP300, GLU68, GLY158, TYR88, ASP155, and ASP87. This combination of non-polar and polar residues highlights a moderate binding mode, largely governed by hydrophobic interactions.

Figure 9E illustrates the docking pose of caryophyllene within the bacterial protein 5YNB. All observed contacts are *van der Waals* in nature, involving residues such as GLY71, GLY73, ASP75, ASN101, LEU100, and TYR132. The absence of stronger interactions (e.g., hydrogen bonds or π interactions) suggests a lower binding affinity and a relatively passive accommodation within the binding cavity.

Figure 9F shows the interaction with the 5CPO protein. Caryophyllene forms *alkyl* interactions with VAL407 and PHE405, alongside *van der Waals* contacts involving ILE681, VAL391, GLN332, ASN337, and PRO387, among others. The diversity of surrounding residues indicates a flexible and largely hydrophobic binding site, suitable for the bulky, cyclic nature of the ligand.

Figure 9G highlights caryophyllene’s interactions with the 7Z4S protein. Notable contacts include *van der Waals* and *alkyl* interactions with VAL297, VAL303, and PHE294, in addition to a potential *pi–sigma* interaction involving PHE294. These hydrophobic and aromatic interactions may support stable binding within the protein’s functional domain, although experimental studies are needed to confirm this.

**Table 14 viruses-17-00951-t014:** Summary of caryophyllene–protein interactions.

Figure	Protein (PDB ID)	Key Interactions	Notable Residues	Binding Nature
9A	7NIO (main protease)	Pi–alkyl, van der Waals	LEU405, ARG560, ASP534	Hydrophobic
9B	5X29 (QS receptor)	Pi–alkyl, van der Waals	PHE28, LEU19, SER16	Hydrophobic
9C	7B3O (resistance)	Alkyl, van der Waals	LEU455, TYR453, ARG403	Hydrophobic
9D	2GTP	Alkyl, van der Waals	LEU227, VAL160, ASP300	Mixed hydrophobic
9E	5YNB	Van der Waals only	GLY71, ASP130, TYR132	Weak/non-specific
9F	5CPO	Alkyl, van der Waals	VAL407, PHE405, ILE681	Hydrophobic
9G	7Z4S	Alkyl, van der Waals, π-σ	VAL297, PHE294, ARG298	Aromatic–hydrophobic

This analysis underscores that caryophyllene preferentially binds to hydrophobic and semi-polar regions of diverse proteins through non-covalent interactions, predominantly alkyl and van der Waals forces. Its lack of hydrogen bonding is consistent with its lipophilic terpenoid structure, suggesting that its biological activity may derive from steric interference with active or regulatory sites in proteins.

The comparative ADMET profile presented in Table 15 offers a comprehensive evaluation of methyl *eugenol* and β-*caryophyllene* using predictive in silico models, emphasizing key pharmacokinetic and toxicological parameters essential for drug development and safety assessment.

In terms of absorption, eugenol displays a markedly higher aqueous solubility than caryophyllene (log S = −2.671 vs. −5.555), which favors its dissolution and potential oral bioavailability. Both compounds show comparable Caco-2 permeability and high human intestinal absorption, suggesting effective gastrointestinal uptake. Despite their favorable absorption profiles, both exhibit relatively low skin permeability, with eugenol being slightly less permeable dermally. Notably, neither molecule is a substrate or inhibitor of P-glycoproteins, indicating a reduced risk of efflux-related drug interactions and enhanced systemic retention.

Regarding distribution, caryophyllene shows a higher volume of distribution (log VDss = 0.652) compared to eugenol (0.265), suggesting more extensive tissue penetration. Both compounds exhibit moderate plasma protein binding, ensuring an adequate unbound drug fraction. Additionally, caryophyllene demonstrates a greater capacity to permeate the blood–brain barrier, though both show limited CNS permeability, which may restrict their central effects.

Under metabolic profiling, neither compound is a substrate or inhibitor of major cytochrome P450 enzymes except for CYP1A2, where eugenol acts as an inhibitor. This interaction implies a potential for metabolic interference when co-administered with substrates of this enzyme. Otherwise, their metabolic stability appears favorable, with low risk of enzymatic inhibition-based drug interactions.

In the context of excretion, caryophyllene exhibits higher predicted total clearance, suggesting faster systemic elimination. Neither compound is likely to be excreted via renal OCT2-mediated pathways.

Concerning toxicity, a notable distinction arises in AMES test predictions: eugenol may possess mutagenic potential, whereas caryophyllene is predicted to be non-mutagenic. Eugenol, however, is tolerated at a higher dose in humans and demonstrates lower acute and chronic toxicity in rodent models. Both compounds show potential for skin sensitization. Environmentally, eugenol exhibits higher minnow toxicity, while caryophyllene is more toxic to *T. pyriformis*, indicating species-specific ecological risks.

In summary, eugenol and caryophyllene share several favorable ADMET traits, such as high absorption and low hepatic toxicity, but differ in solubility, metabolic enzyme interaction, and certain toxicity endpoints. These differences could influence their suitability in pharmaceutical or nutraceutical applications, depending on the target profile and safety requirements.

Molecular Dynamics Simulation of Methyl Eugenol Bound to Key SARS-CoV-2 Proteins

A suite of molecular dynamics (MD) simulations was conducted to evaluate the dynamic stability, structural adaptability, and interaction profiles of methyl eugenol in complex with seven critical SARS-CoV-2 target proteins: 7NIO (nucleocapsid phosphoprotein), 5X29 (Envelope Protein), 7B3O (RNA-dependent RNA polymerase), 2GTH (main protease), 5YNB (RNA-binding domain), 5CPO (papain-like protease), and 7Z4S (spike receptor-binding domain). The simulations were carried out over a 50 ns timescale under explicit solvent conditions, with systematic evaluation of key structural descriptors, including root mean square deviation (RMSD), root mean square fluctuation (RMSF), radius of gyration (Rg), molecular and polar surface area (MSA/PSA), solvent-accessible surface area (SASA), radial distribution function (RDF), and ligand–protein contact profiles as shown in Appendix A, Figure 10, and Table 16.

RMSD and Conformational Equilibrium

Across all protein complexes, RMSD values stabilized within the first 5–6 ns, with most systems maintaining deviations below 2.5 Å. This included the 7Z4S and 7NIO complexes, which exhibited particularly low RMSD plateaus around 1.8–2.1 Å, underscoring rapid convergence and minimal global rearrangement. These results reflect strong structural resilience upon ligand binding, with methyl eugenol being effectively accommodated in shallow grooves, hydrophobic pockets, or active site clefts without inducing deleterious conformational changes. The 7B3O and 5CPO systems similarly stabilized early, with average RMSD values suggesting tight ligand anchoring and thermodynamic compatibility.

RMSF and Residue-Level Mobility

The RMSF analyses revealed that loop regions and termini consistently displayed the highest fluctuations (>2.0 Å), while structured α-helices and β-strands remained notably rigid (≤1.2 Å). Binding-site residues in all complexes showed suppressed mobility, particularly in the 7Z4S RBD, where ACE2-interacting loops exhibited reduced flexibility, hinting at potential inhibitory effects through rigidification. In the catalytic proteins (5CPO, 2GTH), ligand-induced dampening of flexibility in enzymatically active residues supports the hypothesis of functional interference. The 5YNB complex, involving the RNA-binding domain, displayed ligand-induced stabilization within its β-sheet core, aligning with the notion of conformational anchoring.

Radius of Gyration (Rg) and Tertiary Structure Stability

All protein systems maintained compact tertiary structures throughout the simulations, with Rg fluctuations confined within ±0.3 Å of their initial values. No evidence of large-scale unfolding or domain collapse was observed, even in structurally sensitive systems like 2GTH and 5CPO. This compactness confirms the biophysical compatibility of methyl eugenol with diverse viral protein architectures and further supports its non-denaturing interaction profile.

MSA, PSA, and Surface Adaptation

The MSA and PSA trends were consistent across simulations, with moderate increases during early equilibration phases, reflecting solvent reorganization and mild residue rearrangements. PSA elevations (typically ~300 Å^2^) were most notable in proteins with solvent-facing polar grooves (e.g., 5X29, 7B3O), suggesting polar residue reorientation around the ligand. In the 7Z4S system, a slight reduction in MSA indicated hydrophobic cavity occupation by methyl eugenol, leading to reduced surface entropy and enthalpic stabilization.

RDF and Interaction Specificity

RDF profiles across all complexes exhibited sharp, high-density peaks between 2.2 and 4.8 Å, confirming persistent and spatially localized interactions. Notably, 5CPO and 7Z4S showed strong RDF peaks corresponding to active-site residues and receptor-binding motifs, respectively, indicating sustained contact zones critical for function modulation. The narrowness and symmetry of RDF peaks in all systems confirmed single, well-defined binding poses, with no evidence of alternative orientations or transient dissociation events.

SASA and Solvent Stability

The SASA trajectories across complexes revealed minor oscillations consistent with protein breathing motions. Importantly, no spikes indicative of partial unfolding were observed. Slight SASA reductions in the vicinity of the ligand (e.g., 5CPO and 7Z4S) reflected hydrophobic shielding and potential reductions in conformational entropy—features correlated with enhanced binding affinity and reduced solvation energy penalties.

Ligand–Protein Interaction

Methyl eugenol exhibited diverse, yet consistent interaction profiles, dominated by hydrophobic contacts, hydrogen bonds, and π-π or π–cation interactions. In catalytic proteins such as 5CPO and 2GTH, key residues like Cys111 and His272 engaged in stabilizing interactions with the ligand, suggesting active-site targeting. In the 7Z4S RBD, aromatic residues (Tyr453, Tyr505) formed π-π stacking interactions, while Gln493 and Asn487 provided hydrogen bonds—highlighting a pharmacophoric alignment potentially disruptive to ACE2 engagement. The interaction persistence across simulations supports the ligand’s high residence time and specificity.

Finally, methyl eugenol demonstrates a robust and adaptable binding profile across several critical SARS-CoV-2 proteins, maintaining structural stability and inducing local conformational reinforcement at functional domains. Its capacity to stably occupy enzymatic clefts, receptor interfaces, and RNA-binding grooves, while preserving protein fold and compactness, highlights its promise as a phytochemical scaffold for antiviral intervention. These molecular dynamics simulations offer compelling evidence for its further exploration through in vitro validation, SAR optimization, and structure-based drug design initiatives targeting viral replication and host-cell entry.

Comprehensive Molecular Dynamics Simulation of Caryophyllene Bound to Key SARS-CoV-2 Proteins

A series of molecular dynamics (MD) simulations were conducted to assess the dynamic behavior and structural stability of caryophyllene when bound to seven critical SARS-CoV-2 proteins: nucleocapsid phosphoprotein (7NIO), Envelope Protein (5X29), RNA-dependent RNA polymerase (7B3O), main protease (2GTH), RNA-binding domain (5YNB), papain-like protease (5CPO), and the spike receptor-binding domain (7Z4S). Each protein–ligand complex was simulated over a 50 ns timescale, and key structural descriptors were evaluated to understand the conformational response, interaction patterns, and overall compatibility of the non-polar sesquiterpene ligand, as shown in Appendix A, Figure 11, and Table 17.

Stability and Compactness: RMSD, Rg, and SASA Analyses

Across all complexes, RMSD values stabilized within the first 4–6 ns, with average deviations ranging from ~1.8 Å (7Z4S) to ~2.4 Å (5X29), indicating high structural fidelity upon ligand binding. No global unfolding or large-scale structural rearrangements were observed. The radius of gyration (Rg) profiles remained within a narrow ±0.3 Å band, further underscoring the preservation of protein compactness. Similarly, SASA trends demonstrated minimal deviation, suggesting that ligand binding did not provoke abnormal solvent exposure or collapse of tertiary structures.

Local Flexibility and Functional Regions: RMSF Insights

RMSF analyses showed generally low residue-level fluctuations (<1.5 Å) across the α-helical, β-sheet, and catalytic cores. Notably, loop regions in RNA-binding or receptor-interacting domains displayed slightly elevated mobility, yet this remained within expected dynamic limits. In many cases (e.g., 7B3O, 2GTH, 5YNB), ligand presence actually dampened flexibility near active or binding sites, suggesting a stabilizing influence on functionally relevant residues.

Surface and Solvation Behavior: MSA, PSA, and RDF Observations

Molecular surface area (MSA) increased modestly in most systems due to accommodation of the ligand’s hydrophobic volume. However, polar surface area (PSA) remained largely stable, indicating that critical polar interactions and solvation interfaces were conserved. Radial distribution function (RDF) analyses consistently revealed sharp peaks between 4.3 and 4.8 Å, reflecting organized hydration shells that supported ligand stability via water-mediated van der Waals and dispersive forces.

Interaction Mechanisms: Ligand–Protein Contact Analysis

Despite lacking polar functional groups, caryophyllene consistently engaged in van der Waals, alkyl–alkyl, and π–alkyl interactions across all protein targets. Key residues involved included Leu50 and Val73 in 7NIO, Leu28 and Phe56 in 5X29, Ile548 and Phe793 in 7B3O, Met49 and His163 in 2GTH, Phe55 and Ile89 in 5YNB, Leu162 and Val266 in 5CPO, and Tyr489 and Leu455 in 7Z4S. These hydrophobic contacts formed stable clusters that anchored the ligand within shallow grooves, clefts, or surface pockets, without disrupting polar interaction networks or essential catalytic motifs.

Functional Implications and Structural Impact

The simulation data suggest that caryophyllene maintains protein structural integrity while inducing localized rigidification in flexible or catalytically important regions. In spike RBD (7Z4S), reduced mobility in the receptor-binding loop implies potential interference with ACE2 docking. In proteases (2GTH, 5CPO), occupancy near the catalytic triad hints at possible non-competitive inhibition. For RNA-interacting proteins (7NIO, 5YNB), the ligand may modulate RNA accessibility without obstructing functional core motifs.

Finally, the molecular dynamics simulations across seven SARS-CoV-2 proteins demonstrated that caryophyllene binding is structurally non-disruptive and thermodynamically favorable. The ligand’s hydrophobic nature enabled stable anchoring through non-polar contacts, while the preservation of compactness, surface integrity, and functional domain architecture suggests strong compatibility. These findings advocate for the potential of caryophyllene as a scaffold for further antiviral drug development, particularly in targeting protein surfaces involved in viral assembly, RNA binding, or host interaction.

## 4. Discussion

Medicinal and natural plants have been utilized for centuries in combating infectious diseases, including viral infections. They are valued for their wide array of biologically active compounds that can act at multiple stages of the viral cycle. Compounds such as flavonoids, terpenes, alkaloids, and phenolics can disrupt viral attachment, inhibit replication enzymes, or interfere with viral protein synthesis. This multifaceted activity makes plant-based extracts promising tools for the development of antiviral therapies, especially against resistant or emerging viruses [31]. The present study provides comprehensive pharmacogenomic and molecular insight into the antiviral potential of *Syzygium aromaticum* bioactives—primarily eugenol and caryophyllene—against SARS-CoV-2, integrating GC-MS profiling, molecular docking, pharmacokinetic modeling, and miRNA-21 rs1292037 genotyping. These findings build on prior knowledge of phytotherapeutic agents by combining host genetics and viral inhibition in a unified therapeutic framework.

The GC-MS analysis revealed a rich composition of volatile compounds, with caryophyllene (47.55%), methyl eugenol (17.75%), and eugenyl acetate (14.85%) as the major constituents. Caryophyllene, a sesquiterpene hydrocarbon, has documented anti-inflammatory and antiviral activity, partly through modulation of the CB2 cannabinoid receptor and the NF-κB pathway [32,33]. Methyl eugenol and eugenyl acetate contribute not only to aroma but also to antioxidant and cytoprotective effects, which could support host immunity [34]. While eugenol comprised a relatively smaller fraction (4.21%), its broad-spectrum bioactivity, including antibacterial, antifungal, and antiviral properties, is well-established [35,36].

Molecular docking results indicated that both compounds interact with multiple SARS-CoV-2 proteins involved in replication, immune evasion, and viral maturation. Eugenol displayed stronger binding to the Envelope Protein (5X29), Spike RBD (7B3O), and NSP15 (2GTH), indicating its potential to interfere with viral entry and endoribonuclease activity. These findings align with studies by Aboubakr et al. (2021) and Islam et al. (2021), which demonstrated similar docking behavior for eugenol [37,38]. Caryophyllene, in contrast, showed greater binding affinity to the viral RNA polymerase (NSP12, 5CPO) and methyltransferase (NSP16, 5YNB), suggesting that it may inhibit replication and RNA capping post-entry. These complementary interactions support the rationale for a multi-target strategy against SARS-CoV-2.

Furthermore, docking interaction maps confirmed that eugenol forms hydrogen bonds, π-π stacking, and van der Waals contacts with key amino acids across different protein targets. These interactions may enhance ligand stability and specificity, as described in prior structural studies [8]. Caryophyllene, being hydrophobic, formed predominantly alkyl and van der Waals interactions, consistent with its non-polar nature. While this may lead to lower binding energies, it suggests a mechanism involving steric hindrance or membrane interaction rather than active-site inhibition [39].

The ADMET profiling supports the potential of both compounds as safe and orally bioavailable agents. Eugenol displayed superior solubility and a favorable intestinal absorption profile, though its potential mutagenicity in the AMES test warrants caution and dose optimization [40]. Caryophyllene, with its higher tissue distribution and non-mutagenic profile, may offer complementary pharmacokinetic properties. Neither compound showed significant inhibition of cytochrome P450 enzymes (except CYP1A2 for eugenol), reducing the likelihood of metabolic drug–drug interactions. These properties collectively highlight their suitability for repurposing as antiviral agents or for formulation into adjunctive therapies.

The molecular dynamics simulations conducted for methyl eugenol and caryophyllene across seven SARS-CoV-2 target proteins—7NIO, 5X29, 7B3O, 2GTH, 5YNB, 5CPO, and 7Z4S—reveal significant insights into their binding stability, dynamic behavior, and structural influence on protein–ligand complexes.

For *methyl eugenol*, most complexes demonstrated early RMSD fluctuations that settled into stable trajectories, particularly with the 7NIO, 5X29, and 7Z4S proteins, indicating strong conformational retention and equilibrium. The RMSF values, especially for 7NIO and 7B3O, exposed localized flexibility at loops and termini, while the protein cores remained rigid. Radius of gyration (Rg) measurements confirmed the preservation of protein compactness, with minimal expansion throughout the simulations. Both MSA and PSA exhibited minor oscillations, suggesting biological adaptability without destabilization. RDF peaks at short atomic distances in all cases signified consistent atomic-scale interactions, and the stability of SASA readings across simulations further confirmed the exposure and persistence of methyl eugenol at the binding interfaces.

Notably, surface area metrics (MSA and PSA) showed controlled fluctuation across all systems, reinforcing the notion that caryophyllene does not significantly distort protein topology. RDF plots further revealed distinct binding events at proximity levels typically associated with van der Waals and hydrophobic interactions. SASA remained consistent across all simulations, supporting the stable solvation of the ligand–protein interfaces.

When comparing both ligands, *caryophyllene demonstrated a slight edge in complex stability*, particularly with 7Z4S and 5CPO, where the lowest residue fluctuations and sustained spatial compactness were observed. On the other hand, methyl eugenol showed stronger atomic proximity in the RDF plots for proteins like 2GTH and 7NIO, possibly due to its smaller molecular size and better fit within tighter pockets. Nonetheless, both ligands showed broad-spectrum interaction potential, evidenced by stable binding across multiple viral targets.

Overall, the *7Z4S–caryophyllene complex* emerged as the most dynamically stable among all combinations tested, marked by early structural convergence, minimal surface disruption, and preserved protein compactness. These findings highlight caryophyllene’s potential as a lead antiviral scaffold and justify further experimental exploration.

The miR-21 rs1292037 polymorphism analysis added a critical host genetic dimension. Our results demonstrated a significant association between genotype frequency and COVID-19 susceptibility. The TC genotype was more prevalent among infected individuals, while the CC genotype appeared protective. This is consistent with findings from Zhang et al. (2021) and Li et al. (2022), who reported altered miRNA expression profiles in severe COVID-19 cases [41,42]. As miR-21 regulates inflammatory pathways—including TGF-β, PTEN/PI3K-AKT, and NF-κB—its altered function due to SNP variation could significantly modulate cytokine expression.

Indeed, the elevated IL-6, D-dimer, and ferritin levels observed in the TT and TC genotypes suggest a link between the polymorphism and hyperinflammatory states. Prior research has correlated miR-21 upregulation with IL-6-driven cytokine storms and endothelial damage in COVID-19 [43,44]. Furthermore, the positive correlation of IL-6 with IL-13 and D-dimer in our study reflects the concurrent activation of Th2 cytokines and coagulation pathways, which have been previously implicated in disease progression and mortality [45,46].

Sex- and age-specific analyses provided further insight. Male patients exhibited higher inflammatory marker levels, consistent with known immunological differences and sex hormone influences on cytokine regulation [47]. Older patients showed increased disease severity, likely reflecting immune senescence, as reported by Akbar and Gilroy (2020) [48].

The integration of phytochemical profiling, molecular docking, and SNP genotyping provides a multifactorial approach to understanding antiviral resistance and host response. From a translational perspective, these data support the use of *Syzygium aromaticum* extracts—particularly eugenol and caryophyllene—in targeted therapeutics, especially in genetically predisposed individuals. However, further in vitro and clinical validation is essential before these agents can be recommended as standalone antivirals or immunomodulators.

## 5. Conclusions

This study presents an integrative analysis of the antiviral potential of *Syzygium aromaticum* bioactives—particularly eugenol and caryophyllene—against SARS-CoV-2, combining GC-MS profiling, molecular docking, pharmacokinetic modeling, and host genetic analysis via miR-21 rs1292037 genotyping. Both compounds demonstrated strong binding affinities toward critical viral proteins implicated in entry, replication, and RNA processing, supporting a multi-target antiviral mechanism. The methoxy-substituted eugenol derivative, in particular, exhibited favorable interactions with NSP15, NSP16, and the Envelope Protein, underscoring its potential as a promising therapeutic scaffold. ADMET evaluations further indicated that these phytochemicals possess suitable pharmacokinetic and safety profiles, although careful dose optimization remains necessary for eugenol. Caryophyllene and methyl eugenol both demonstrated stable binding with SARS-CoV-2 target proteins in molecular dynamics simulations. Among all complexes, the caryophyllene–7Z4S complex showed the highest structural stability, suggesting caryophyllene as a promising antiviral candidate. From a host genetic perspective, elevated levels of IL-6, IL-10, IL-13, MMP-7, ferritin, and D-dimer were observed in COVID-19 patients across all genotypes, with the CC genotype associated with the highest concentrations, suggesting a heightened inflammatory and pro-thrombotic state. These findings implicate the miR-21 rs1292037>C polymorphism as a critical modulator of immune and coagulation responses during SARS-CoV-2 infection. The association between the CC genotype and increased inflammatory burden positions this SNP as a potential biomarker for disease susceptibility and severity. Thus, genotype-informed risk stratification may enhance clinical decision-making and therapeutic targeting. Collectively, these insights highlight the dual importance of phytochemical intervention and host genetic profiling in the development of personalized antiviral strategies. Future in vitro and in vivo investigations are essential to validate these findings and translate them into clinical applications.

Availability of data and materials: Appendix A illustrate the molecular dynamics simulations of methyl eugenol and caryophyllene in complex with SARS-CoV-2 target proteins.

## Figures and Tables

**Figure 1 viruses-17-00951-f001:**
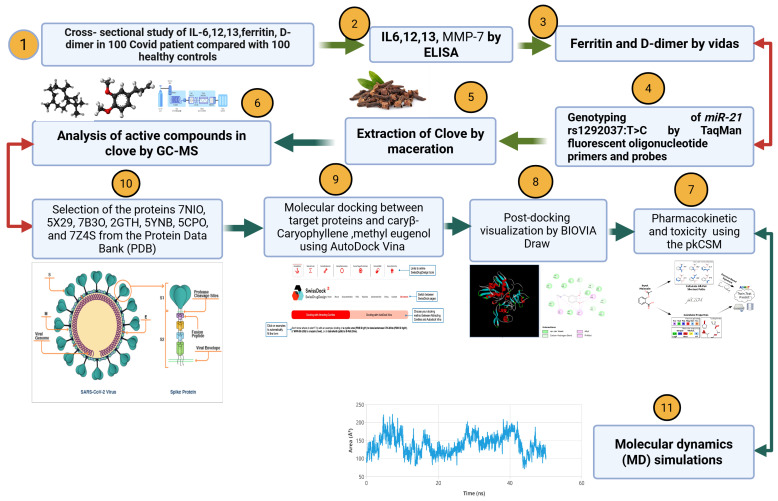
Comprehensive clinical and computational framework investigating clove phytochemicals as potential therapeutics against COVID-19. This figure presents an integrative approach combining clinical biomarker analysis, phytochemical profiling, and computational modeling to explore the antiviral potential of Syzygium aromaticum compounds. Key steps include cytokine and genetic profiling in COVID-19 patients, GC-MS analysis of clove extracts, pharmacokinetic evaluation, molecular docking with viral and resistance-related proteins, and validation through molecular dynamics simulations. This figure was designed by the researcher Mohammed Mukhles Ahmed using the BioRender platform under a paid subscription, https://app.biorender.com/illustrations/685e788053a15de0a8c031cb (accessed on 1 July 2025).

**Figure 2 viruses-17-00951-f002:**
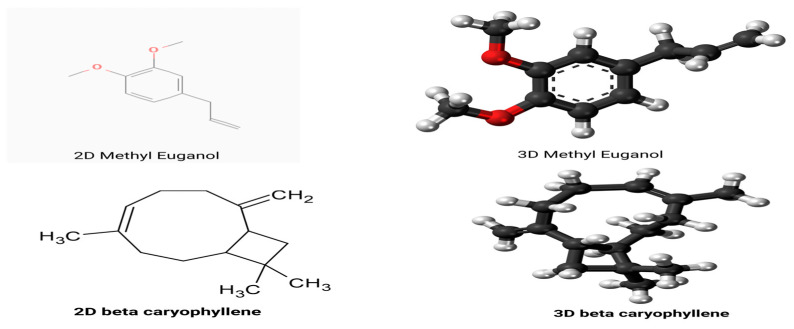
Two-dimensional and three-dimensional structure of methyl eugenol β-Caryophyllene.

**Figure 3 viruses-17-00951-f003:**
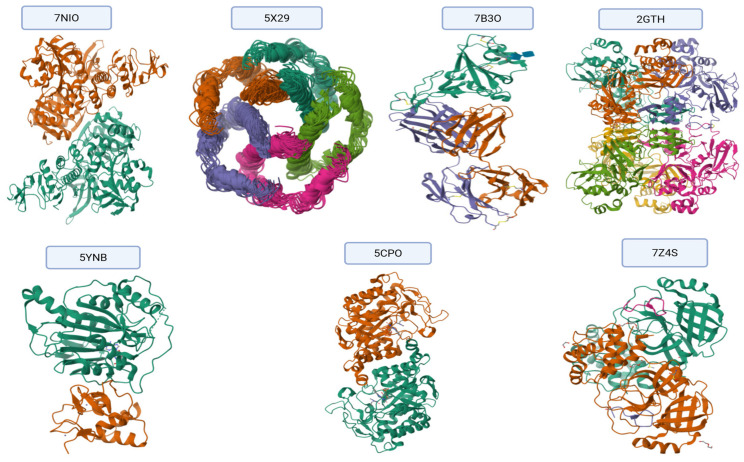
Selected proteins of COVID-19 were obtained from the Protein Data Bank (PDB) https://www.rcsb.org [24]. Each color represents an individual polypeptide chain or structural domain for visual clarity.

**Figure 4 viruses-17-00951-f004:**
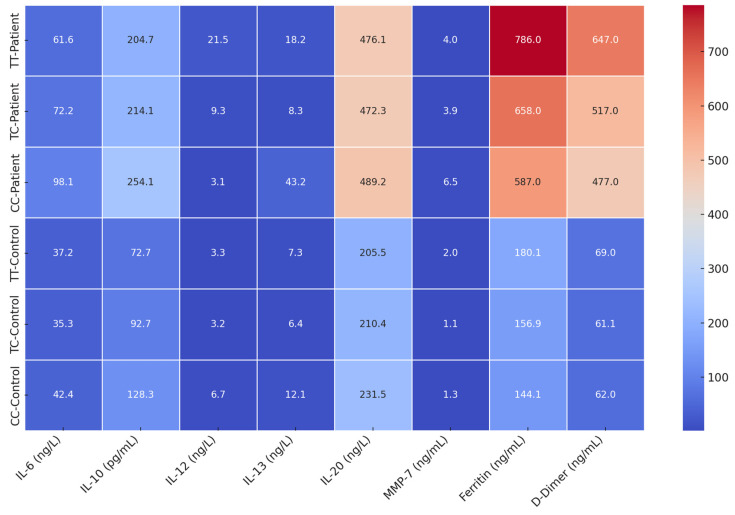
Heatmap showing cytokine and coagulation levels across genotypes in patients and controls. Heatmap showing the concentration levels of inflammatory and biochemical markers (IL-6, IL-10, IL-12, IL-13, IL-20, MMP-7, Ferritin, and D-Dimer) across different study groups: CC-Control, TC-Control, Ce-Control, TC-Patient, and TP-Patient. The color gradient reflects the magnitude of expression, with higher concentrations indicated by warmer colors.

**Figure 5 viruses-17-00951-f005:**
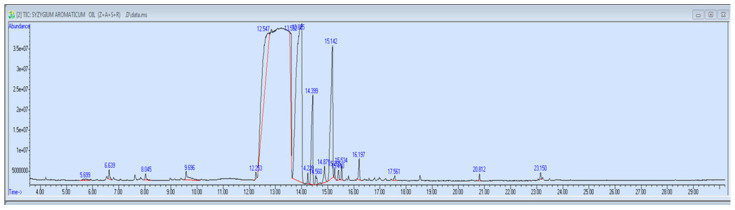
GC-MS of active compounds for clove extract.

**Figure 6 viruses-17-00951-f006:**
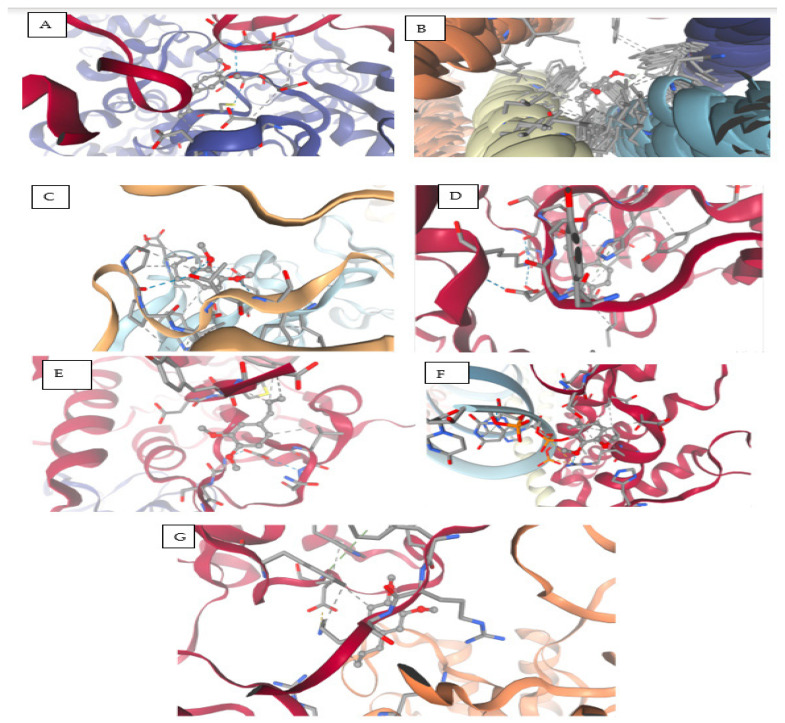
Interaction of ligand COC1=C(C=C(C=C1)CC=C)OC with multiple SARS-CoV-2 targets. (**A**): 7nio_modified.pdb; (**B**): Envelope (E) Protein (5X29); (**C**): Spike (S) Protein RBD (7b3o); (**D**): NSP15 (2gth); (**E**): 5ynb_modified.pdb; (**F**): 9cpo; (**G**): 7z4s.

**Figure 7 viruses-17-00951-f007:**
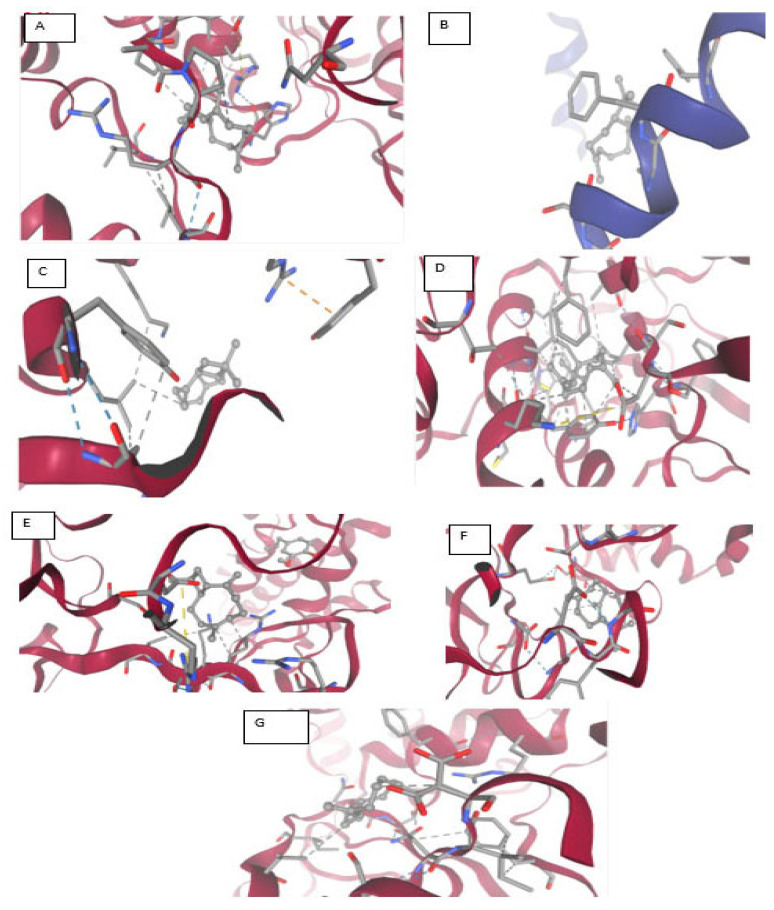
Interaction of ligand C/C/1=C\CCC(=C)[C@H]2CC([C@@H]2CC1)(C)C) with multiple SARS-CoV-2 targets. (**A**): 7nio_modified.pdb; (**B**): Envelope (E) Protein (5X29); (**C**): Spike (S) Protein RBD (7b3o); (**D**): NSP15 (2gth); (**E**): 5ynb_modified.pdb; (**F**): 9cpo; (**G**): 7z4s.

**Figure 8 viruses-17-00951-f008:**
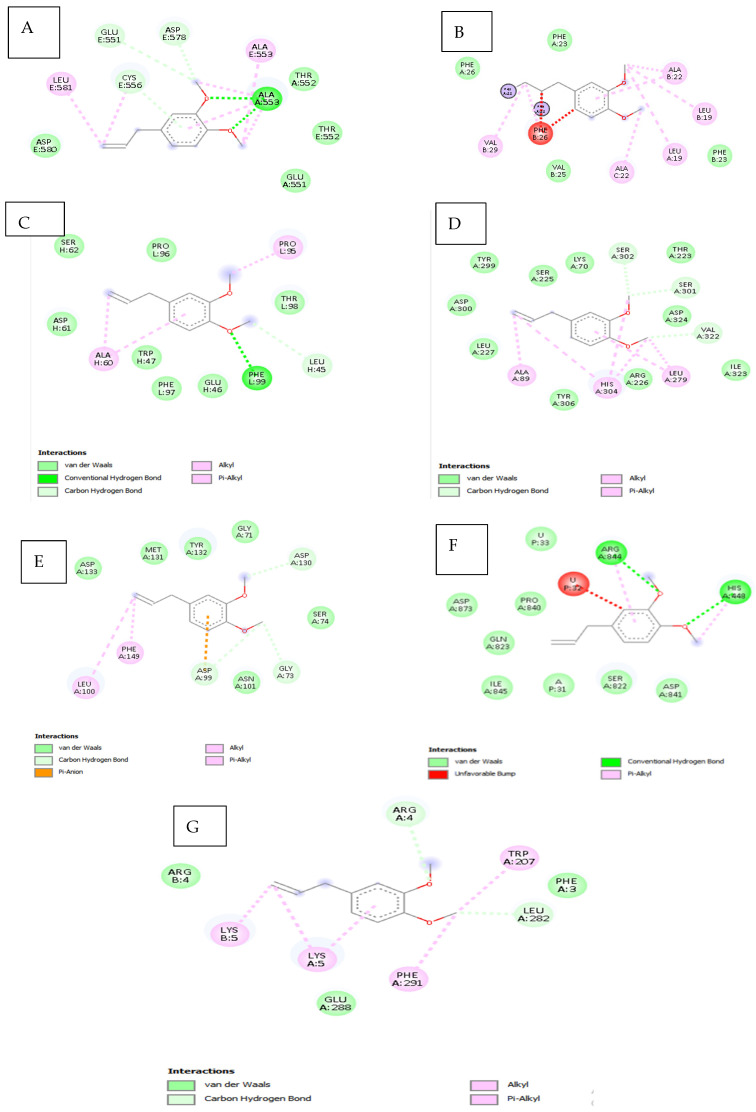
Interaction of ligand COC1=C(C=C(C=C1)CC=C)OC with multiple SARS-CoV-2 targets (2D). (**A**): 7nio_modified.pdb; (**B**): Envelope (E) Protein (5X29); (**C**): Spike (S) Protein RBD (7b3o); (**D**): NSP15 (2gth); (**E**): 5ynb_modified.pdb; (**F**): 9cpo; (**G**): 7z4s.

**Figure 9 viruses-17-00951-f009:**
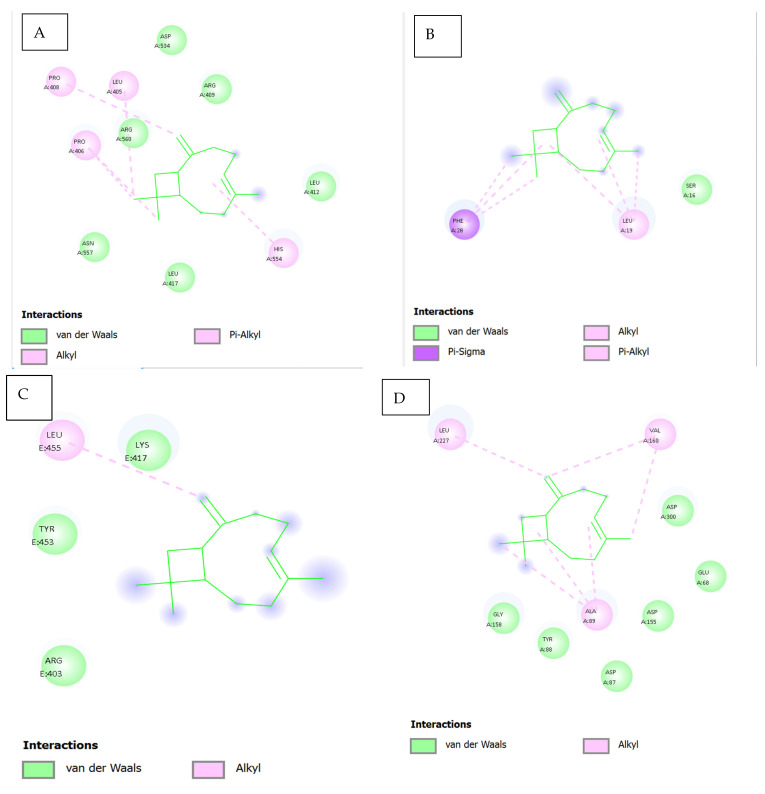
Interaction of ligand C/C/1=C\CCC(=C)[C@H]2CC([C@@H]2CC1)(C)C with multiple SARS-CoV-2 targets (2D). (**A**): 7nio_modified.pdb; (**B**): Envelope (E) Protein (5X29); (**C**): Spike (S) Protein RBD (7b3o); (**D**): NSP15 (2gth); (**E**): 5ynb_modified.pdb; (**F**): 9cpo; (**G**): 7z4s.

**Figure 10 viruses-17-00951-f010:**
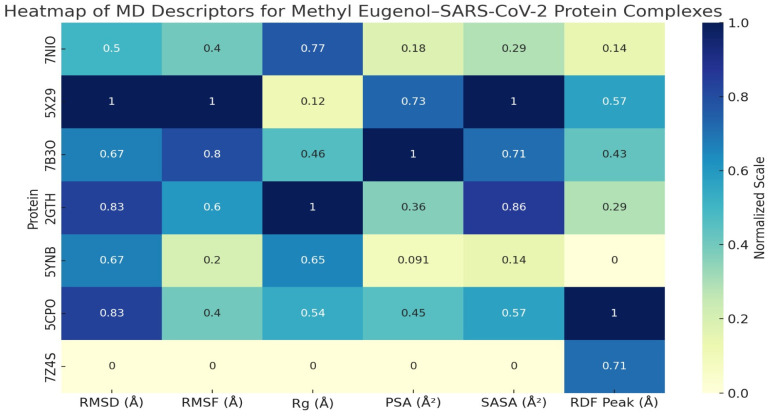
Heat map of molecular dynamics parameters for SARS-CoV-2 protein complexes with methyl eugenol.

**Figure 11 viruses-17-00951-f011:**
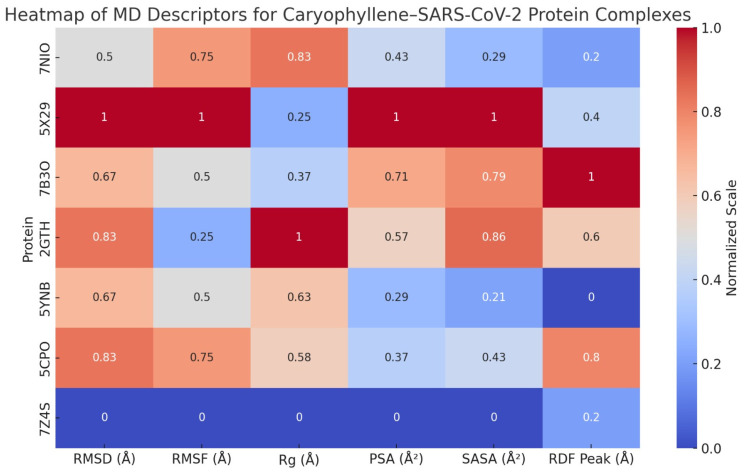
Heat map of molecular dynamics parameters for SARS-CoV-2 protein complexes with β-caryophyllene.

**Table 1 viruses-17-00951-t001:** Primer and probe sequences for the studied SNPs.

SNPs	Primers	Probes
rs1292037:T>C	Forward: ATGGAGGGAGGATTTTATGGAGAA	rs1292037P/C: FamAAGCTGCACTGTGGGT
	Reverse: GAAGGTCAAGTAACAGTCATACAGC	rs1292037-P/T: Hex-ACTAAGCTGCATTGTGGGT
Annealing temp.: 60 °C. The mutation data are already available on the NCBI https://pubmed.ncbi.nlm.nih.gov/34791386/ (accessed on 1 January 2022)		

**Table 2 viruses-17-00951-t002:** Target proteins of COVID-19.

No.	Protein Name	Full Name	Function	Justification for Targeting	PDB ID
1	NSP5 (Mpro)	Main protease (3CLpro)	Cleaves viral polyprotein	Inhibition may block viral protein maturation	7Z4S
2	NSP12	RNA-dependent RNA polymerase (RdRp)	Viral RNA synthesis	Target for replication inhibition	9CPO
3	NSP16	2′-O-methyltransferase	Viral mRNA capping	Prevents immune evasion	5YNB
4	NSP15	Endoribonuclease (NendoU)	RNA processing	Disrupts immune escape	2GTH
5	Spike Protein (RBD)	Receptor-binding domain	Host cell entry	Blocking ACE2 interaction	7B3O
6	NSP13	RNA helicase	Unwinds viral RNA	Interrupts genome replication	7NIO
7	Envelope (E) Protein	Structural Envelope Protein	Viral assembly and release	Reduces virion formation	5X29

**Table 3 viruses-17-00951-t003:** Characterization of docking for selected target proteins.

No.	Target Protein	PDB ID	Grid Center Coordinates (x, y, z)
1	NSP15 (Endoribonuclease)	2GTH	(17.108, −63.576, −78.744)
2	NSP13 (RNA Helicase)	7NIO	−17.958, 19.866, −27.020)
3	Envelope Protein (E)	5X29	(−11.008, −6.004, 7.319)
4	Spike Protein (RBD)	7B3O	(−58.000, −9.000, 27.000)
5	NSP16 (2′-O-Methyltransferase)	5YNB	(63.000, 86.000, 147.000)
6	NSP12 (RNA-dependent RNA Polymerase)	5CPO	(146.000, 122.000, 122.000)
7	NSP5 (Main Protease, Mpro)	7Z4S	(13.000, 9.000, −4.000)

**Table 4 viruses-17-00951-t004:** Demographic characteristics of the study population.

Variable	COVID-19 Patients (*n* = 100)	Controls (*n* = 100)
Sex		
Male	60 (60%)	54 (54%)
Female	40 (40%)	46 (46%)
Age Group		
<40 years	39 (39%)	47 (47%)
≥40 years	61 (61%)	53 (53%)
BMI Classification		
Underweight	0 (0%)	0 (0%)
Normal Weight	42 (35 M, 7 F)	34 (15 M, 19 F)
Overweight	34 (24 M, 10 F)	40 (23 M, 17 F)
Obese	24 (4 M, 20 F)	26 (15 M, 11 F)

**Table 5 viruses-17-00951-t005:** Cytokine and coagulation marker levels.

Marker	Patients (Mean ± SD)	Controls (Mean ± SD)	*p*-Value
IL-6 (ng/L)	77.3 ± 7.3	38.3 ± 5.5	≤0.01
IL-12 (ng/L)	11.3 ± 1.1	4.4 ± 0.4	≤0.01
IL-13 (ng/L)	23.2 ± 1.5	8.6 ± 0.7	≤0.01
MMP-7 (ng/mL)	4.8 ± 0.5	1.5 ± 0.1	≤0.01
Ferritin (ng/mL)	677 ± 107	160.4 ± 20.7	≤0.01
D-Dimer (ng/mL)	547 ± 67.9	64.0 ± 13.3	≤0.01

**Table 6 viruses-17-00951-t006:** Sex-based differences in ferritin and D-dimer.

Marker	Gender	Patients (Mean ± SD)	Controls (Mean ± SD)	*p*-Value
Ferritin (ng/mL)	Male	594.0 ± 34.9	56.2 ± 9.6	0.00003
	Female	476.9 ± 37.9	74.2 ± 10.4	<0.0001
D-Dimer (ng/mL)	Male	742.2 ± 69.4	166.4 ± 21.7	<0.0001
	Female	581.3 ± 79.0	152.5 ± 17.0	0.0001

**Table 7 viruses-17-00951-t007:** Correlation coefficients (Pearson’s r) among markers.

Biomarker Pair	Correlation (r)	Strength
IL-6 vs. IL-12	0.719	Strong
IL-6 vs. IL-13	0.787	Strong
IL-12 vs. D-Dimer	0.847	Strong
Ferritin vs. D-Dimer	0.796	Strong
IL-13 vs. Ferritin	0.386	Weak

**Table 8 viruses-17-00951-t008:** Cytokine and coagulation profiles by genotype in COVID-19 patients.

Genotype	IL-6	IL-10	IL-12	IL-13	IL-20	MMP-7	Ferritin	D-Dimer
TT	61.6	204.7	21.5	18.2	476.1	4.0	786	647
TC	72.2	214.1	9.3	8.3	472.3	3.9	658	517
CC	98.1	254.1	3.1	43.2	489.2	6.5	587	477

**Table 9 viruses-17-00951-t009:** Cytokine and coagulation profiles by genotype in healthy controls.

Genotype	IL-6	IL-10	IL-12	IL-13	IL-20	MMP-7	Ferritin	D-Dimer
TT	37.2	72.7	3.3	7.3	205.5	1.98	180.1	69.0
TC	35.3	92.7	3.2	6.4	210.4	1.14	156.9	61.1
CC	42.4	128.3	6.7	12.1	231.5	1.26	144.1	62.0

**Table 10 viruses-17-00951-t010:** *t*-test results for biomarkers (patients vs. controls).

Biomarker	T-Statistic	*p*-Value	Significance
IL-6	3.531	0.06436	Not Significant
IL-10	5.688	0.00479	Significant
IL-12	1.249	0.32912	Not Significant
IL-13	1.389	0.29291	Not Significant
IL-20	27.792	0.00004	Significant

**Table 11 viruses-17-00951-t011:** Compounds identified in clove extract via GC-MS.

Peak	Retention Time (min)	Compound Name(s)	Area %	CAS Number(s)
1	5.699	β-Pinene, Bicyclo[3.1.1]heptane, 6,6-dimethyl-2-methylene	0.30	000127-91-3/018172-67-3
2	6.642	Acetic acid, dec-2-yl ester; 5-methylhex-2-yl ester	0.46	1000367-89-6/others
3	8.044	1,5-Heptadiene, 3,3-dimethyl-, (E); 3,6-dimethyl-; esters	0.37	067682-47-7/others
4	9.697	Methyl salicylate	1.21	000119-36-8
5	12.251	α-Cubebene, α-Copaene	0.30	017699-14-8/others
6	12.545	Methyl Eugenol	17.75	000097-53-0
7	13.592	Eugenol, Phenol, 2-methoxy-3-(2-propenyl)-	4.21	000097-53-0/001941-12-4
8	13.887	β-Caryophyllene, Bicyclo[5.2.0]nonane derivative	47.55	000087-44-5/242794-76-9
9	14.242	trans-Isoeugenol, Eugenol, 3-Allyl-6-methoxyphenol	0.41	005932-68-3/others
10	14.397	1,4,7-Cycloundecatriene, Humulene	7.20	006753-98-6/1000062-61-9
11	14.562	γ-Muurolene, other terpenes	0.70	030021-74-0/others
12	14.873	α-Farnesene	1.18	000502-61-4
13	15.142	Eugenyl acetate, Eugenol	14.85	000093-28-7/000097-53-0
14	15.263	trans-Calamenene, other hydrocarbons	0.36	1000374-17-3/others
15	15.410	Naphthalene derivative, α-Cubebene	0.57	016728-99-7/017699-14-8
16	15.531	Benzenamine, 2-bromo-6-chloro-4-nitro-, quinazoline derivative	0.61	000099-29-6/others
17	16.198	Caryophyllene oxide	1.03	001139-30-6
18	17.565	2′,3′,4′-Trimethoxyacetophenone	0.31	013909-73-4
19	20.811	Hexadecanoic acid, methyl ester	0.31	000112-39-0
20	23.148	9-Octadecenoic acid, methyl ester (E)	0.34	001937-62-8

**Table 12 viruses-17-00951-t012:** Comparative binding affinities of eugenol and caryophyllene with SARS-CoV-2 target proteins using AutoDock Vina.

No.	Protein (Function)	PDB ID	Role in Viral Life Cycle	Eugenol Binding Affinity (kcal/mol)	Caryophyllene Binding Affinity (kcal/mol)
1	NSP13 (Helicase)	7NIO	Unwinding of viral RNA	−4.599	−4.082
2	Envelope (E) Protein	5X29	Virus assembly, budding, and pathogenesis	−5.267	−3.573
3	Spike (S) Protein—RBD	7B3O	ACE2 receptor binding; viral entry	−4.979	−1.152
4	NSP15 (Endoribonuclease)	2GTH	Viral RNA processing; immune evasion	−5.317	−4.588
5	NSP16 (2′-O-Methyltransferase)	5YNB	RNA cap methylation; immune evasion	−5.170	−6.000
6	NSP12 (RNA-dependent RNA Polymerase)	5CPO	Viral RNA replication	−4.415	−6.200
7	NSP5 (Main Protease—3CLpro)	7Z4S	Cleavage of polyproteins; viral maturation	−4.638	−5.500

**Table 13 viruses-17-00951-t013:** Interaction of (COC1=C(C=C(C=C1)CC=C)OC) with amino acid residues among SARS-CoV-2 targets.

Target PDB ID	Residue (Chain:Position)	Interaction Type	Bond/Feature Description	Remarks
7nio	ALA (A:553/E:553)	Hydrogen bond/hydrophobic	Green dashed/pink dashed	Polar + non-polar interaction
	THR (A:552/E:552)	Polar contact	Nearby, not bonded	Within interactive range
	GLU (A:551/E:551)	Electrostatic/polar	Salt bridge or hydrogen bond	Charge-based stabilization
	CYS (E:556)	Van der Waals/polar	Green dashed	Binding stabilization
	ASP (E:578/E:580)	Electrostatic/polar	Green dashed/nearby	Ionic/secondary interactions
	LEU (E:581)	Hydrophobic	Pink dashed	Hydrophobic pocket formation
5X29	PHE (A/B:25–26), A:23	Pi–Pi stacking/hydrophobic	Red dashed/π-interaction	Strong aromatic contact
	ALA, LEU, VAL (A/B/C:19–29)	Hydrophobic/van der Waals	Pink dashed/nearby	Non-polar stabilizing contacts
7b3o	PHE (L:99), LEU (H:45), ALA (H:60), PRO (L:95)	H-bond/C–H bond/Pi–alkyl	Directional H-bond/Pi-electron cloud	Key anchoring interactions
	SER, ASP, TRP, GLU, THR (Various)	Van der Waals	Weak non-specific	Minor stabilization
2gth	HIS:304, SER:301	C–H/Pi–alkyl	Light gray/light purple	Moderate interaction
	TYR, ASP, LEU, SER, LYS, ILE, ARG, VAL (Various)	Van der Waals	Green	Hydrophobic + spatial fitting
	ALA:89, LEU:279	Alkyl	Purple	Stabilizing hydrophobic bonds
5ynb	ASP:A:99,130,133	Pi–anion/C–H bond/van der Waals	Orange/light green/solid green	Electrostatic + hydrogen bonding
	ASN:A:101, GLY:A:71,73, MET:A:131, TYR:A:132	Mixed polar/VdW	Various	Complex surface binding
	LEU:A:100, PHE:A:149	Pi-Alkyl	Pink dashed	Aromatic and hydrophobic pocket
9cpo	ARG:A:844, HIS:A:448	Hydrogen bond/Pi–alkyl	Green/pink dashed	Strong and stabilizing contacts
	U:P:32–33, A:P:31	Van der Waals/unfavorable bump	Solid green/red dashed	Possible steric hindrance
	ASP, PRO, GLN, ILE, SER (Various)	Van der Waals	Solid green	Non-specific interactions
7z4s	ARG:A/B:4, LEU:A:282	Carbon–hydrogen bond	Light green dashed	Weak polar interaction
	LYS:A/B:5, TRP:A:207, PHE:A:291	Pi–alkyl	Pink dashed	Aromatic hydrophobic anchoring
	GLU:A:288, PHE:A:3	Van der Waals	Solid green circle	Additional stabilizers

**Table 15 viruses-17-00951-t015:** Comparative ADMET profiles of eugenol and caryophyllene predicted using in silico models.

Category	Model Parameter	Eugenol	Caryophyllene	Unit/Type	Interpretation
Absorption	Water Solubility	−2.671	−5.555	log mol/L	Eugenol exhibits greater aqueous solubility than caryophyllene, enhancing its bioavailability potential.
	Caco-2 Permeability	1.528	1.423	log Papp (10^−6^ cm/s)	Both compounds demonstrate moderate permeability across intestinal epithelial cells.
	Intestinal Absorption (Human)	94.532	94.845	% Absorbed	Both compounds show high predicted human intestinal absorption.
	Skin Permeability	−1.916	−1.58	log Kp	Eugenol shows lower skin permeability than caryophyllene, indicating weaker dermal penetration.
	P-glycoprotein Substrate	No	No	Yes/No	Neither compound is a P-gp substrate, reducing the risk of efflux-mediated bioavailability loss.
	P-glycoprotein I Inhibitor	No	No	Yes/No	No inhibitory interaction with P-gp I, suggesting low transporter-based drug interaction risk.
	P-glycoprotein II Inhibitor	No	No	Yes/No	Similarly, both compounds do not inhibit P-gp II.
Distribution	Volume of Distribution (VDss, Human)	0.265	0.652	log L/kg	Caryophyllene shows wider distribution throughout body tissues than eugenol.
	Fraction Unbound (Human)	0.208	0.263	Fu	Both exhibit moderate plasma protein binding, allowing a reasonable free fraction in circulation.
	Blood–Brain Barrier (BBB) Permeability	0.422	0.733	log BB	Caryophyllene has a higher potential to cross the BBB compared to eugenol.
	CNS Permeability	−1.922	−2.172	log PS	Both compounds demonstrate poor permeability into CNS tissue.
Metabolism	CYP2D6 Substrate	No	No	Yes/No	Neither compound is a substrate for CYP2D6, reducing metabolic variability.
	CYP3A4 Substrate	No	No	Yes/No	Neither is a substrate for CYP3A4, suggesting limited metabolism via this major enzyme.
	CYP1A2 Inhibitor	Yes	No	Yes/No	Eugenol may inhibit CYP1A2, potentially affecting the metabolism of co-administered drugs.
	CYP2C19 Inhibitor	No	No	Yes/No	Both show no inhibition of CYP2C19.
	CYP2C9 Inhibitor	No	No	Yes/No	No interaction with CYP2C9 predicted for either compound.
	CYP2D6 Inhibitor	No	No	Yes/No	Neither compound is a CYP2D6 inhibitor.
	CYP3A4 Inhibitor	No	No	Yes/No	Neither compound interferes with CYP3A4-mediated drug metabolism.
Excretion	Total Clearance	0.338	1.088	log mL/min/kg	Caryophyllene demonstrates a higher clearance rate, possibly due to enhanced metabolic or renal elimination.
	Renal OCT2 Substrate	No	No	Yes/No	Neither compound is eliminated via OCT2-mediated renal secretion.
Toxicity	AMES Toxicity	Yes	No	Yes/No	Eugenol may exhibit mutagenic potential; caryophyllene does not.
	Max. Tolerated Dose (Human)	0.959	0.351	log mg/kg/day	Eugenol has a higher predicted safe daily dose in humans.
	hERG I Inhibitor	No	No	Yes/No	Neither compound shows cardiotoxic risk via hERG I inhibition.
	hERG II Inhibitor	No	No	Yes/No	Both are non-inhibitors of hERG II channels.
	Oral Rat Acute Toxicity (LD_50_)	1.973	1.617	mol/kg	Eugenol is predicted to be less acutely toxic than caryophyllene in rats.
	Oral Rat Chronic Toxicity (LOAEL)	2.119	1.416	log mg/kg_bw/day	Eugenol shows a higher threshold for chronic toxicity than caryophyllene.
	Hepatotoxicity	No	No	Yes/No	Neither compound is predicted to cause liver damage.
	Skin Sensitization	Yes	Yes	Yes/No	Both compounds may provoke allergic skin responses upon dermal exposure.
	*T. pyriformis* Toxicity	0.742	1.401	log µg/L	Caryophyllene exhibits greater toxicity toward *T. pyriformis*.
	Minnow Toxicity	1.449	0.504	log mM	Eugenol shows higher aquatic toxicity than caryophyllene.

**Table 16 viruses-17-00951-t016:** Comparative molecular dynamics analysis of methyl eugenol binding to key SARS-CoV-2 proteins.

Protein (PDB ID)	Avg. RMSD (Å)	RMSF Insight	Rg Stability	SASA Behavior	Key Interactions	Functional Implication
7NIO	~2.1	Suppressed flexibility in RNA-binding loops	±0.3 Å	Stable, no unfolding	Hydrophobic and van der Waals contacts	Stabilizes RNA recognition domain
5X29	~2.4	Flexible polar loops, stable transmembrane region	±0.3 Å	Minor fluctuations from surface reorganization	Polar groove reorganization	Preserves Envelope Protein topology
7B3O	~2.2	Reduced mobility in catalytic core	±0.3 Å	Stable with slight oscillations	π–π stacking with aromatic residues	Rigidifies polymerase catalytic cleft
2GTH	~2.2	Dampened motion in active site residues	±0.3 Å	Stable, hydrophobic shielding near ligand	Cys145, His41 (active-site)	May inhibit protease activity
5YNB	~2.0	Stabilized β-sheet core	±0.3 Å	Maintained solvent exposure	Anchoring near RNA-binding surface	Anchors RNA-binding domain
5CPO	~2.2	Suppressed catalytic triad mobility	±0.3 Å	Hydrophobic surface shielding	Cys111, His272 (active-site)	Suggests allosteric inhibition
7Z4S	~1.8	Rigidified ACE2-interacting loops	±0.3 Å	Reduced SASA near RBD	Tyr453, Gln493, π–π and H-bonds	Disrupts ACE2-binding via loop rigidity

**Table 17 viruses-17-00951-t017:** Comparative molecular dynamics analysis of caryophyllene binding to Key SARS-CoV-2 proteins.

Protein (PDB ID)	Avg. RMSD (Å)	RMSF Insight	Rg Stability	SASA Behavior	Key Interactions	Functional Implication
7NIO	~2.1	Stable core, flexible RNA-binding loops	±0.3 Å	Stable, no unfolding	Leu50, Val73 (hydrophobic)	Stabilizes RNA-binding groove
5X29	~2.4	Stable transmembrane region, flexible polar loops	±0.3 Å	Minor loop exposure	Leu28, Phe56 (hydrophobic)	Preserves envelope topology
7B3O	~2.3	Stable catalytic motifs, dampened binding site	±0.2 Å	Consistent solvent accessibility	Ile548, Phe793 (hydrophobic)	Rigidifies polymerase active site
2GTH	~2.2	Catalytic dyad stabilized, flexible distal loops	±0.3 Å	No significant change	Met49, His163 (hydrophobic)	Potential protease inhibition
5YNB	~2.0	Rigid β-sheet core, reduced loop mobility	±0.2 Å	Maintained exposure pattern	Phe55, Ile89 (hydrophobic)	May modulate RNA access
5CPO	~2.2	Catalytic triad stable, flexible peripheral loops	±0.3 Å	Steady, with loop breathing	Leu162, Val266 (hydrophobic)	Allosteric interference possible
7Z4S	~1.8	Rigid core, reduced mobility in receptor loop	±0.2 Å	Slight reduction due to compaction	Tyr489, Leu455 (hydrophobic)	Blocks ACE2-binding loop flexibility

## Data Availability

The original contributions presented in this study are included in the article/Appendix A.

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
