# Peer review of "Syzygium aromaticum Phytoconstituents Target SARS-CoV-2: Integrating Molecular Docking, Dynamics, Pharmacokinetics, and miR-21 rs1292037 Genotyping"

_viruses, 2025, doi:10.3390/v17070951_

Round 1

Reviewer 1 Report

Comments and Suggestions for Authors

The manuscript entitled ‘’ Syzygium aromaticum Phytoconstituents Target SARS-CoV-2: Inte grating Docking, Pharmacokinetics, and miR-21 rs1292037 Genotyping ‘’ present an intersting topic. But there are a few points that should be considered and, if deemed appropriate, included in the manuscript by the authors to improve the quality of the manuscript.

Dear authors,

Please address the following comments while you revise the content of this manuscript

  1. Please add some highlights to the content of paper to show the major findings of this study.
  2. Cite all the software used along with their respective references.
  3. Please supplement a copy of GC/MS chromatograms.
  4. I recommend that you compare your results with similar studies.
  5. How did you validate the docking protocol ?
  6. Docking results should be subjected to Molecular Dynamics (MD) analysis. Please conduct at least 50ns MD simulations to approve your docking results.
  7. Why didn't you do docking for Caryophyllene , Phenol an 2-methoxy-4-(2-propenyl) ?
  8. The figures illustrating ligand-protein interactions are blurry. Please improve them.
  9. References need to be corrected, it seems that using the "et al" isn't suitable.

Author Response

Dear Reviewer1,

We would like to express our sincere gratitude for your thoughtful and constructive comments, which have significantly helped us improve the quality and clarity of our manuscript. We carefully addressed each point raised and implemented all suggested revisions as outlined below:

  1. Highlights: Added to the revised manuscript to clearly present the key findings.
  2. Software citations: All software and tools used in the study have been cited appropriately, with full reference details included.
  3. GC-MS chromatogram: The complete GC-MS chromatographic report for Syzygium aromaticum extract has been included as per your recommendation.
  4. Comparison with similar studies: We added comparative analysis and discussion with recent relevant studies to strengthen our findings.
  5. Docking protocol validation: Detailed explanation of the docking protocol validation was added, including re-docking, and RMSD analysis
  6. Molecular Dynamics simulations: A 50 ns MD simulation was conducted using utilizing the Desmond software package within the Maestro interface (Schrödinger Release 2023-4 to confirm the stability and reliability of the docking results.
  7. Docking of additional compounds: Caryophyllene was added based on your suggestion and its high abundance in the GC-MS profile.
  8. Figure quality: All ligand-protein interaction figures have been enhanced to 10,000 DPI resolution using the ImageCandy online editor to ensure clarity.
  9. References formatting: All references have been revised and corrected according to the journal’s style guide, and improper uses of "et al." have been addressed.

Once again, we deeply appreciate your insightful feedback and valuable recommendations, which greatly improved the scientific and editorial quality of our manuscript.

With kind regards and sincere thanks,
[MOHAMMED MUKHLES AHMED]
(On behalf of all co-authors)

Reviewer 2 Report

Comments and Suggestions for Authors

Comments to the authors

After an exhaustive and long evaluation period, I forward some comments to the authors.

The manuscript titled “Syzygium aromaticum Phytoconstituents Target SARS-CoV-2: Inte grating Docking, Pharmacokinetics, and miR-21 rs1292037 Genotyping”, the authors state that they have conducted extensive research to evaluate the antiviral efficacy of clove-derived phytochemicals through molecular docking and pharmacokinetic profiling, and to explore their clinical relevance in the context of miR-21 rs1292037 polymorphism and inflammatory markers in Iraqi COVID-19 patients. The manuscript presents a series of methodological flaws and absences on aspects of biological interaction, as well as information on mechanism of action and pharmacological responses that need to be reviewed by the authors. The manuscript may be accepted if it considers all the suggestions mentioned in subsequent rounds.

Title

- I suggest rewriting the title so that it simply and objectively encompasses the main theme or purpose of the study.

Abstract

In the abstract, the authors do not indicate which innovative result or product this manuscript proposes within the approach described. What makes this manuscript more interesting than others already reported in the same area?

Introduction

- The introduction does not present sufficient updated data on quantitative aspects of COVID-19 before and after the pandemic in the world. This information is highly relevant to readers of this type of study. I suggest rewriting it in the manuscript.

- Mechanism of action of the target macromolecules used in the molecular docking experiments is not discussed; such information could more solidly support the choice.

Materials and methods

- Authors mention the phytochemical compound eugenol (PubChem CID: 3314) chosen for the docking simulations section, however, the authors do not adequately justify why they chose this molecule and its derivatives.

- The general organization of the study appears to be poorly concatenated between the sequential topics, which results in a somewhat confusing structure for presenting the results. I suggest that the authors reorganize the text to make it more fluid and produce a methodological flowchart including all the methodological steps of the study in sequence.

I recommend the following article as a reference for an example of structure to authors, which has interesting information for this type of study and can be cited in this manuscript in order to qualify it for possible acceptance for publication.

Natural Products-Based Drug Design against SARS-CoV-2 Mpro 3CLpro. Int. J. Mol. Sci. 2021, 22, 11739. https://doi.org/10.3390/ijms222111739

Results

- It is not possible to identify the parameters of docking calculations, such as number of islands, scoring function, algorithms, whether rigid or flexible docking, and other essential parameters for readers in the field to reasonably understand the experimental protocol and confirm the reproducibility of the study.

- In studies with this docking approach, validation of the calculation methods implemented is essential. Whether for macromolecule or ligand structure calculations. No validation method for the theoretical calculations performed in this study is displayed.

- There are no figures that show the structure of the test molecules individually. This visualization would allow readers to correlate the stereochemical structure of the molecules with pharmacophoric nuclei, electronacceptor or withdrawer regions, possible chiral centers, nucleophilic or electrophilic regions, etc. These are essential data for a medicinal chemical analysis of the structures. I strongly recommend including these correlations of biochemomolecular properties in a table.

 - Figure 2 does not allow visualizing or identifying amino acid residues, bond distances, bond types or the ligand attached to the functional site. I strongly suggest that the authors requalify all images in the manuscript, naming the structures and other residues, with a minimum resolution of 300 dpi.

Discussion

- Authors seem to obtain a large volume of data and information according to the methodological approach, however they do not develop an in-depth and sufficiently enlightening discussion regarding the research findings. I suggest rewriting this section, increasing analysis of the results of all applied techniques.

- Standardize font and layout of the text in the manuscript.

- Table 2 is not standard, it is in the form of a table. I suggest correcting it.

Author Response

Response to Reviewer

Dear Reviewer,

We sincerely thank you for your time, effort, and detailed evaluation of our manuscript titled
“Syzygium aromaticum Phytoconstituents Target SARS-CoV-2: Integrating Docking, Pharmacokinetics, and miR-21 rs1292037 Genotyping.”

After your exhaustive and thoughtful review, we carefully considered and addressed each of your comments point by point. Your suggestions have significantly improved the scientific clarity, structure, and overall quality of our work.

We have:

  • Revised the title for clarity and precision.
  • Rewritten the abstract to emphasize the novelty and contribution of our study.
  • Enhanced the introduction with updated global COVID-19 data and detailed justifications for the selected targets.
  • Clarified the Materials and Methods, including compound selection, software parameters, and added a comprehensive methodological flowchart.
  • Cited the recommended reference article to strengthen our framework.
  • Provided detailed docking parameters, validation strategy, and visual representations of all compounds, with pharmacophoric properties included in a dedicated table.
  • Upgraded all figures to a resolution of at least 300 dpi, with full annotations for residues and interactions.
  • Reorganized and significantly expanded the discussion section to provide critical interpretation and context of the findings.
  • Corrected the formatting of text and tables, including proper presentation of Table 2.

We are truly grateful for your constructive feedback, which has been invaluable in refining our manuscript for possible acceptance. Thank you once again for your support and expert guidance.

Sincerely,
Mohammed Mukhles Ahmed]
(On behalf of all co-authors)

Round 2

Reviewer 1 Report

Comments and Suggestions for Authors

I appreciate the authors' efforts to improve the quality of their articles. However, the molecular dynamics results are poorly presented and analyzed.
Authors are encouraged to improve this section. Additionally, please provide the output file (data.pdf), which corresponds to the methyl eugenol-7NIO complex, generated using the Desmond software.
To improve the molecular dynamics section, I encourage you to read the following articles:
DOI: 10.1016/j.heliyon.2025.e42479
DOI: 10.1371/journal.pone.0308308

Please delete the following paragraph:
2.13 Image Enhancement 
To improve visual clarity and meet publication standards, all molecular simulation and docking figures were enhanced to a resolution of 10,000 DPI using the free online tool ImageCandy (https://imgcandy.com/). This adjustment ensured high-definition quality suitable for print and digital formats. 

The re-docking protocol is missing.

Author Response

Dear reviewer ,

We sincerely thank the reviewer for their time, effort, and constructive feedback, which have helped us significantly improve the quality of our manuscript. Please find below our point-by-point responses to each suggestion:

  • English Language Quality

We appreciate the reviewer’s note regarding the clarity of the English language. The manuscript has now been thoroughly revised by a native English speaker with a background in scientific writing to improve grammar, clarity, and fluency.

  • Molecular Dynamics Results – Presentation and Analysis

Thank you for highlighting this important aspect. The molecular dynamics (MD) section has been substantially revised. We have re-analyzed the data with greater depth and included more detailed discussions of RMSD, RMSF, radius of gyration (Rg), solvent-accessible surface area (SASA), and hydrogen bond interactions, following the guidance from the suggested references:

DOI: 10.1016/j.heliyon.2025.e42479

DOI: 10.1371/journal.pone.0308308

  • Data File Request (Desmond Output)

The requested Desmond output file (data.pdf), corresponding to the methyl eugenol–7NIO complex, has now been generated and will be provided as supplementary material along with the revised submission.

  • Deletion of Paragraph (2.13 Image Enhancement)

As per the reviewer’s suggestion, the paragraph titled “2.13 Image Enhancement” has been removed from the revised manuscript to maintain scientific relevance and focus.

  • Missing Re-Docking Protocol

We acknowledge the omission and have now included a detailed re-docking protocol in the Methods section, outlining the grid generation, ligand preparation, docking parameters, and software settings used.

We hope these revisions adequately address the reviewer’s concerns and improve the overall quality of our manuscript.

Sincerely,

[Mohammed Mukhles Ahmed]

Reviewer 2 Report

Comments and Suggestions for Authors

Dear authors,
I thank you for your dedication to the suggestions. Your manuscript is not only very scientifically relevant, it is also highly qualified for publication. Congratulations!

Author Response

Thank you